# Socratic-Zero: Bootstrapping Reasoning via Data-Free Agent Co-evolution

## Abstract

Recent breakthroughs in large language models (LLMs) on reasoning tasks rely heavily on massive, high-quality datasets—typically human-annotated and thus difficult to scale. While data synthesis or distillation offers a promising alternative, existing methods struggle with inconsistent data quality and an inability to dynamically adapt to the evolving capabilities of the model, leading to suboptimal training signals. To address these limitations, we introduce *Socratic-Zero*, a fully autonomous framework that generates high-quality training data from minimal seed examples through the co-evolution of three agents: the *Teacher*, the *Solver*, and the *Generator*. The *Solver* continuously refines its reasoning by learning from preference feedback on both successful and failed trajectories; the *Teacher* adaptively crafts increasingly challenging questions based on the Solver's weaknesses; and the *Generator* distills the Teacher's question-design strategy to enable scalable, high-fidelity curriculum generation. This closed-loop system produces a self-improving curriculum—requiring no pre-existing tasks or labels. Remarkably, starting from only 100 seed questions, our *Socratic-Solver-8B* achieves an average gain of +20.2 percentage points over prior data synthesis methods across seven mathematical reasoning benchmarks (AMC23, AIME24-25, Olympiad, MATH-500, Minerva, and GSM8K), with consistent gains on both Qwen3 and GLM4 series models. Furthermore, when using synthetic data from our Socratic-Generator-32B to train student models, they outperform those trained on data from state-of-the-art (SOTA) commercial LLMs on these benchmarks, including Qwen3-235B-A22B, DeepSeek-V3.1-671B, GPT-5, Gemini-2.5-Pro, Grok-4, and Claude-4.1-Opus.

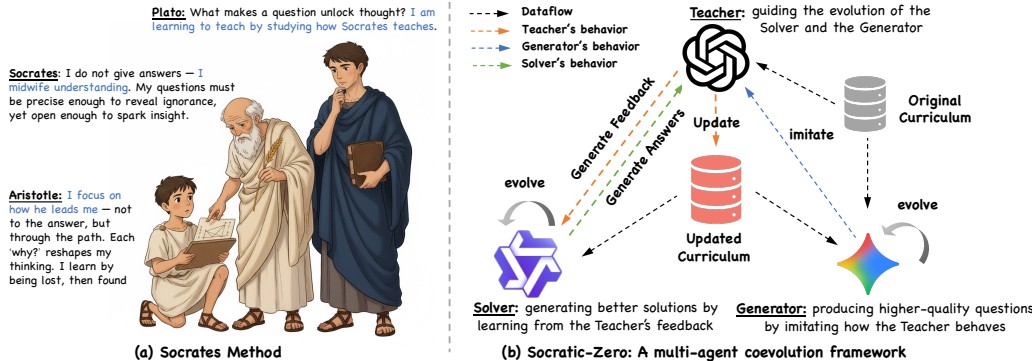

Figure 1: The Socratic-Zero Framework: From Philosophical Analogy to a Co-evolutionary System. **(a) The Socratic Methodlogy** illustrates the philosophical foundation: the **Teacher (Socrates)** acts as an intellectual midwife, eliciting understanding through probing questions; the **Practitioner (Aristotle)** learns not by receiving answers, but by being guided along a path of reasoned inquiry; and the **Apprentice-Teacher (Plato)** learns to teach by observing and internalizing the master's method. **(b) The Socratic-Zero Framework** operationalizes this philosophy. Here, the **Teacher**—a powerful LLM—guides the co-evolution of two agents. The **Solver** improves by generating solutions and refining them through the Teacher's feedback, while the **Generator** evolves by strategically distilling the Teacher's behavior to produce an increasingly suitable curriculum for the Solver.

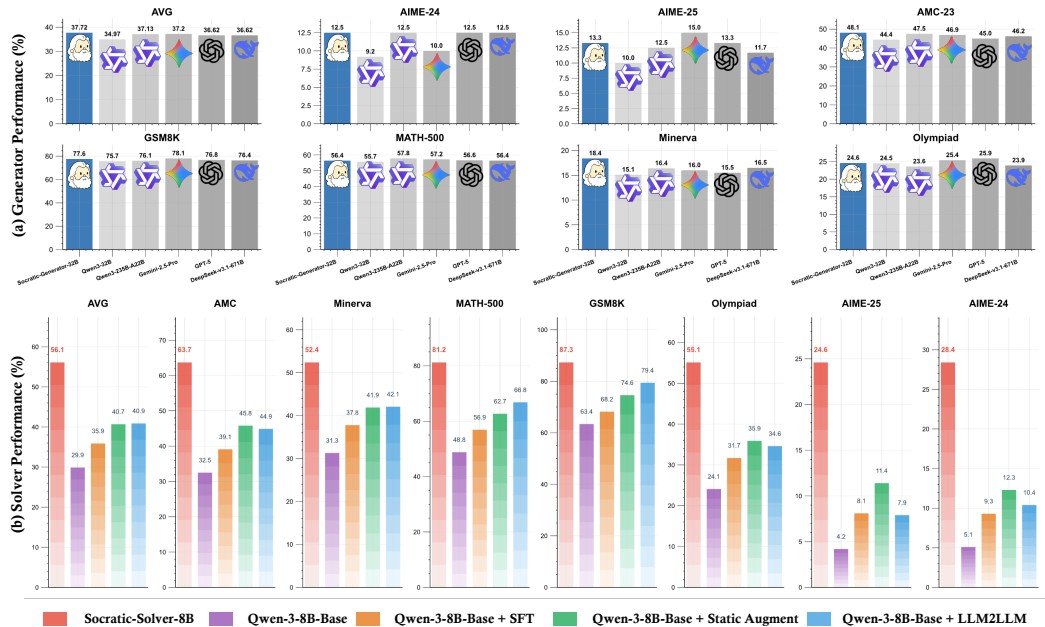

Figure 2: Overall performance comparison demonstrating the giant effectiveness of Socratic-Zero. (a) Our Socratic-Generator-32B produces synthetic data that enables student models to achieve performance competitive with much larger state-of-the-art models, showcasing strong generalization capabilities. (b) Our Socratic-Solver-8B achieves an impressive 56.1% average accuracy, marking a substantial +20.2 point improvement over the baseline.

# 1 INTRODUCTION

The pursuit of advanced mathematical reasoning in large language models has reached a critical juncture. While recent breakthroughs have demonstrated remarkable capabilities on complex mathematical problems (Hendrycks et al., 2021; Cobbe et al., 2021), these advances rely on massive datasets of meticulously curated reasoning trajectories — a requirement that is both costly and fundamentally unscalable. Current state-of-the-art models depend on millions of human-annotated problem-solution pairs and hand-designed curricula (Yu et al., 2024), creating a fundamental bottleneck that limits both accessibility and the potential for models to evolve beyond human-curated knowledge boundaries.

Current methodologies remain entrenched in a static paradigm: datasets are frozen upon collection, curricula are handcrafted in advance, and models are trained on fixed problem distributions. This approach suffers from critical weaknesses: it cannot adapt to evolving model capabilities during training, fails to exploit rich feedback signals for targeting specific weaknesses, and requires extensive human expertise for curriculum design. Recent efforts through synthetic data generation (Lee et al., 2024; Chen et al., 2025b) and iterative training (Zhao et al., 2025a; Huang et al., 2025b) have shown promise but remain constrained by their reliance on external supervision and lack of effective quality control mechanisms for synthesized content.

To overcome these limitations, we introduce **Socratic-Zero**, a paradigm-shifting framework that eliminates dependency on large-scale external datasets while enabling truly autonomous reasoning improvement. Inspired by the Socratic method of learning through questioning (Figure 1(a)), our approach implements co-evolution between three agents: a *Solver* that attempts to solve mathematical questions, a *Teacher* that strategically generates challenging problems to expose the Solver's weaknesses, and a *Generator* that learns to distill and scale the Teacher's problem generation strategy. This architecture (Figure 1(b)) translates the philosophical dialogue of the Socratic method into a concrete, co-evolutionary computational framework. Unlike conventional pipelines that decouple data generation from model training, Socratic-Zero unifies them within a continuous co-evolutionary loop formalized as a optimization problem. Our contributions are threefold:

- **Multi-Agent Co-Evolutionary Framework:** We establish a theoretical foundation for co-evolutionary learning where the Solver, Teacher, and Generator agents interact dynamically, formalizing reasoning improvement as an adaptive curriculum learning problem (Figure 1(b)).
- **Socratic-Zero System:** We implement a concrete framework where the Solver improves via preference learning, the Teacher evaluates correctness and generates adaptive curriculum, and the Generator learns strategic distillation through value-weighted supervised fine-tuning (WSFT), enabling autonomous reasoning advancement from minimal seed data.
- **Superior Empirical Performance:** Our Socratic-Solver-8B achieves +20.2 points average improvement across seven mathematical reasoning benchmarks (Figure 2(b)), while synthetic data from our Socratic-Generator-32B achieves 37.72% downstream training effectiveness, higher than the effectiveness of data generated by leading commercial models including Qwen3-235B-A22B at 37.13%, Claude-4.1-Opus at 37.63%, Gemini-2.5-Pro at 37.20%, Grok-4 at 37.01%, GPT-5 at 36.62%, and DeepSeek-V3.1 at 36.62% (Figure 2(a)).

## 2 RELATED WORK

**Data Synthesis.** To alleviate data scarcity, researchers have leveraged LLMs' generative capabilities to synthesize training samples. Early approaches used prompt engineering to guide question-answer generation (Yu et al., 2024; Zhan et al., 2025). Subsequently, LLM2LLM (Lee et al., 2024) and WarriorMath (Chen et al., 2025b) introduced deficiency-aware mechanisms, where teacher models identify knowledge gaps and generate targeted data. More recently, Absolute Zero (Zhao et al., 2025a) and R-Zero (Huang et al., 2025b) explored fully autonomous self-play paradigms for continuous task generation and learning. While these advances achieve data autonomy, they lack effective quality control mechanisms, resulting in repeated use of low-value samples that severely impact effectiveness.

**Data Distillation.** Knowledge distillation transfers capabilities from powerful teacher models to lighter student models. Early work like Orca (Mukherjee et al., 2023) used imitation learning to replicate teacher reasoning chains. Policy distillation (Wang et al., 2025e) extends this by transferring dynamic decision-making strategies. GKD (Agarwal et al., 2024) enables students to learn from their own sequences using teacher feedback for policy correction. However, students passively accept teacher feedback without evaluating reliability, degrading learning quality when guidance is suboptimal. These methods also rely on static datasets, unable to dynamically adjust content based on students' evolving capabilities. While recent advances (Wang et al., 2025c; Zhao et al., 2023; Zhang et al., 2024b; Chen et al., 2024a; Liu et al., 2025b) promote data-centric optimization, they lack effective quality control and adaptive curriculum generation.

**Preference Learning.** Translating feedback signals into model optimization is central to self-evolution systems. Early approaches like RLHF (Stiennon et al., 2020) trained reward models on human preferences then fine-tuned policies, but this process is complex and unstable. Recent methods like DPO (Rafailov et al., 2023) and RWSFT (Mukherjee et al., 2025) directly optimize preferences, improving efficiency and stability. Combined with self-correction mechanisms like Self-Refine (Madaan et al., 2023), models possess preliminary closed-loop capabilities. Further advances including Self-Evolved Reward Learning (Huang et al., 2025a), Self-Play Fine-Tuning (Chen et al., 2024b), and Self-Play Critic (Chen et al., 2025a) explore autonomous feedback strategies. However, these methods lack unified, co-evolving frameworks for feedback generation and validation.

## 3 METHODOLOGY

### 3.1 THE SOCRATIC-ZERO FRAMEWORK

We introduce **Socratic-Zero**, a fully autonomous, co-evolutionary framework designed to bootstrap mathematical reasoning from a minimal set of seed problems, entirely without external data. As illustrated in Figure 3, the system operates as a closed loop comprising three specialized agents: a *Solver* that learns to reason, a *Teacher* that designs challenging problems, and a *Generator* that distills the Teacher's strategy for scalable curriculum synthesis. At each iteration $t$, the framework maintains a dynamic curriculum $\mathcal{C}_t \subset \mathbb{Q}$, where $\mathbb{Q}$ represents the space of all possible reasoning questions.

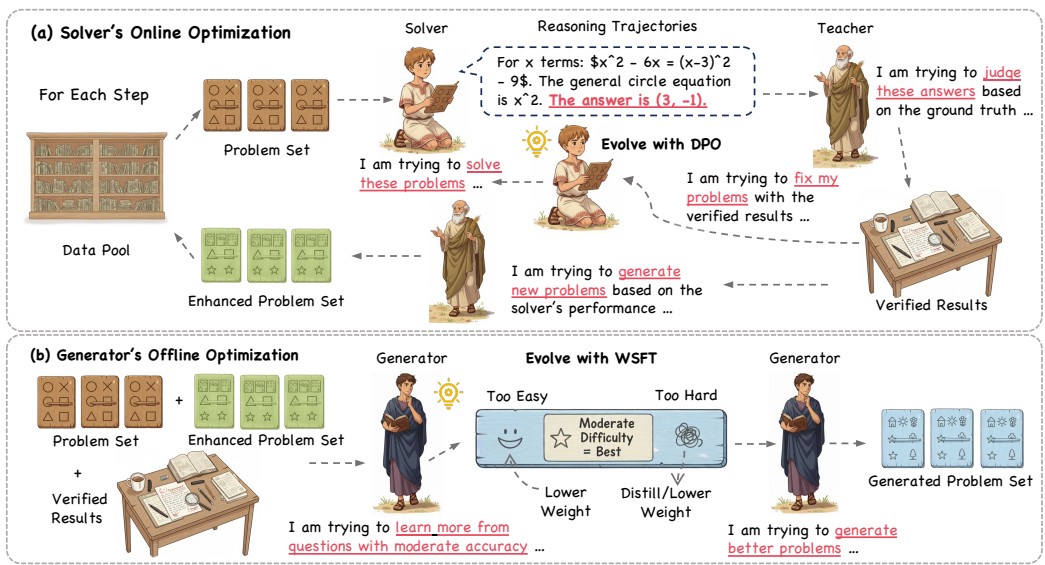

Figure 3: Overview of the Socratic-Zero Framework. (a) Solver Evolving: The Solver attempts to solve problems and learns from preference pairs of correct and incorrect solutions via DPO, while the frozen Teacher strategically generates challenging problems based on Solver failures using fixed generation and evaluation functions. (b) Generator Evolving: The Generator distills the Teacher's problem generation strategy using value-weighted supervised learning. Together, these create a self-improving loop where the curriculum dynamically evolves to maintain optimal challenge levels for the Solver's current capabilities.

The core of Socratic-Zero is the co-evolution of these three agents. The Solver is trained to solve problems from the current curriculum $\mathcal{C}_t$, while the Teacher actively expands $\mathcal{C}_t$ with new problems that are precisely targeted at the Solver's current weaknesses. The Generator, trained in parallel, distills the Teacher's strategy to eventually replace it in later stages of training. This dynamic creates a self-improving loop where the curriculum continuously adapts to the Solver's evolving capabilities, ensuring that the training signals remain maximally informative.

---

### Agents in the Socratic-Zero Framework

1. **Solver ($\mathcal{S}$):** An agent with a policy $\pi_{\mathcal{S}}^{(t)}$ that maps a problem $q$ to a solution trajectory $a_{\mathcal{S}}$. The Solver's objective is to produce correct reasoning steps. It improves by learning from preference feedback on its own attempts, distinguishing its correct solutions from flawed ones.

2. **Teacher ($\mathcal{T}$):** A frozen, high-capacity LLM that serves as a reasoning oracle. It provides two deterministic functions: (i) an evaluation function $\mathcal{T}_{\text{eval}}(q, a_{\mathcal{S}}) \in \{0, 1\}$, which judges the correctness of a Solver's attempt $a_{\mathcal{S}}$ for a problem $q$; and (ii) a problem generation function $\mathcal{T}_{\text{gen}}(q, a_{\mathcal{S}})$, which creates a new problem-solution pair $(q_{\mathcal{T}}, a_{\mathcal{T}})$ by refining an original problem $q$ based on a failed attempt $a_{\mathcal{S}}$.

3. **Generator ($\mathcal{G}$):** An agent with a policy $\pi_{\mathcal{G}}^{(t)}$ that learns to mimic the Teacher's problem-generation strategy. It takes an original problem $q$ and generates a new problem $q_{\mathcal{G}}$. The Generator evolves by learning to produce problems that are optimally challenging for the current Solver, guided by a value function that quantifies a problem's utility.

   - *Teacher-Guided Co-evolution:* The Solver-Teacher and Generator-Teacher interactions occur simultaneously but independently. The Solver learns to solve problems through Teacher evaluation, while the Generator learns to generate problems by imitating the Teacher's curriculum design. The Generator internalizes the Teacher's pedagogical strategies, eventually replace the Teacher and provide more adaptive guidance to the Solver.

   - *Self-Guided Co-evolution:* The Solver-Generator pair co-evolves autonomously without Teacher involvement. The Generator creates adaptive problems tailored to the Solver's current capability, driving mutual improvement through continuous interaction.

The curriculum evolves by incorporating problems generated from the Solver's failures:

$$\mathcal{C}_{t+1} = \mathcal{C}_t \cup \left\{ (q_{\mathcal{T}}, a_{\mathcal{T}}) \,\middle|\, q \in \mathcal{C}_t, \; a_{\mathcal{S}} = \pi_{\mathcal{S}}^{(t)}(q), (q_{\mathcal{T}}, a_{\mathcal{T}}) = \mathcal{T}_{\text{gen}}(q, a_{\mathcal{S}}), \text{s.t.} \, \mathcal{T}_{\text{eval}}(q, a_{\mathcal{S}}) = 0 \right\}, \quad (1)$$

where $\mathcal{C}_0$ is initialized with seed problem-answer pairs.

The Solver and the Generator are co-evolving guided the Teacher. Specifically, on one hand, the Solver minimizes its expected solution loss $\mathcal{L}_{\mathcal{S}}(\mathcal{C}_t, \pi_{\mathcal{S}}^{(t)}(q))$ defined by element-wise loss $\ell_{\mathcal{S}}(a_{\mathcal{S}}, q)$:

$$\mathcal{L}_{\mathcal{S}}(\mathcal{C}_t, \pi_{\mathcal{S}}^{(t)}) = \mathbb{E}_{q \sim \mathcal{C}_t} \big[ \ell_{\mathcal{S}}(a_{\mathcal{S}}, q) \big]. \quad (2)$$

On the other hand, the Generator minimizes its expected generation loss $\mathcal{L}_{\mathcal{G}}(\mathcal{C}_t, \pi_{\mathcal{G}}^{(t)}, v_{\mathcal{G}})$ defined by element-wise loss $\ell_{\mathcal{G}}(q_{\mathcal{G}}, q, q_{\mathcal{T}}, v_{\mathcal{G}})$:

$$\mathcal{L}_{\mathcal{G}}(\mathcal{C}_t, \pi_{\mathcal{G}}^{(t)}, v_{\mathcal{G}}) = \mathbb{E}_{q \sim \mathcal{C}_t} \big[ \ell_{\mathcal{G}}(q_{\mathcal{G}}, q, q_{\mathcal{T}}, v_{\mathcal{G}}) \big]. \quad (3)$$

The full training procedure is summarized in Algorithm 2.

## 3.2 ONLINE SOLVER EVOLUTION VIA PREFERENCE LEARNING

This section elaborates on the evolution of the Solver. The core mechanism is preference learning, where the Solver learns to discriminate between high- and low-quality reasoning using the Teacher's evaluations as the ground truth.

For each problem $q \in \mathcal{C}_t$, the Solver is prompted to generate $k$ distinct solution attempts, $\{a_{\mathcal{S}}^{(i)}\}_{i=1}^k$. These attempts are evaluated by the Teacher's oracle, $\mathcal{T}_{\text{eval}}$, to construct preference pairs. Specifically, for each problem, we partition the solutions into "winning" ($\mathcal{Z}^+$) and "losing" ($\mathcal{Z}^-$) sets:

$$\mathcal{Z}^-(q) = \{a_{\mathcal{S}}^{(i)} \mid \mathcal{T}_{\text{eval}}(q, a_{\mathcal{S}}^{(i)}) = 0\}, \quad (4)$$

$$\mathcal{Z}^+(q) = \begin{cases} \{a_{\mathcal{S}}^{(i)} \mid \mathcal{T}_{\text{eval}}(q, a_{\mathcal{S}}^{(i)}) = 1\} & \text{if } z_q > 0, \\ \{a_{\mathcal{T}}\} & \text{if } z_q = 0, \end{cases} \quad (5)$$

where $z_q = \sum_{i=1}^k \mathcal{T}_{\text{eval}}(q, a_{\mathcal{S}}^{(i)})$ is the number of successful attempts. If all $k$ attempts fail ($z_q = 0$), a single expert solution $a_{\mathcal{T}}$, provided by the Teacher, serves as the sole winning example.

The preference dataset, $\mathcal{D}_{\text{pref}}$, is then constructed by aggregating pairs of $(a^+, a^-)$ where $a^+ \in \mathcal{Z}^+(q)$ and $a^- \in \mathcal{Z}^-(q)$ for all problems in the training batch.

The Solver, parameterized by $\theta_{\mathcal{S}}$, is updated using Direct Preference Optimization (DPO) (Rafailov et al., 2023) to minimize the following loss:

$$\mathcal{L}_{\text{DPO}}(\theta_{\mathcal{S}}; \theta_{\text{ref}}) = -\mathbb{E}_{q \sim \mathcal{C}_t, (a^+, a^-) \sim \mathcal{D}_{\text{pref}}} \left[ \log \sigma \left( \beta \log \frac{\pi_{\theta_{\mathcal{S}}}(a^+|q)}{\pi_{\theta_{\text{ref}}}(a^+|q)} - \beta \log \frac{\pi_{\theta_{\mathcal{S}}}(a^-|q)}{\pi_{\theta_{\text{ref}}}(a^-|q)} \right) \right], \quad (6)$$

where $\pi_{\theta_{\mathcal{S}}}$ is the current Solver policy, $\pi_{\theta_{\text{ref}}}$ is a frozen reference policy (typically the Solver's policy from the previous iteration), $\beta$ is a temperature hyperparameter, and $\sigma$ is the sigmoid function.

## 3.3 OFFLINE GENERATOR EVOLUTION VIA WEIGHTED DISTILLATION

To create a scalable curriculum generation process that does not perpetually depend on the expensive Teacher, the Generator is trained to distill the Teacher's problem-design strategy. The key insight is that an effective training problem is one of *desirable difficulty*—it should be challenging enough to be informative but not so difficult that it is unsolvable for the current Solver.

We quantify this notion using a value function $v(q')$ that scores a problem $q'$ based on the current Solver's success rate. After the Solver makes $k$ attempts on $q'$, resulting in a success count of $z_{q'} = \sum_{i=1}^k \mathcal{T}_{\text{eval}}(q', a_{\mathcal{S}}^{(i)})$, the problem's value is calculated as:

$$v(q') = \exp\left( -\frac{(z_{q'}/k - \mu)^2}{2\sigma^2} \right), \quad (7)$$

where we set the target success rate $\mu = 0.5$ to encourage problems at the frontier of the Solver's ability, and $\sigma$ controls the tolerance for deviation from this ideal difficulty. This unnormalized Gaussian form assigns the highest value to problems solved approximately half the time.

The Generator is trained via weighted supervised fine-tuning (WSFT) to mimic the Teacher. The training data consists of tuples $(q, q_{\mathcal{T}}, v(q_{\mathcal{T}}))$ derived from Solver failures, where $q$ is the original problem, $q_{\mathcal{T}}$ is the Teacher's refined version, and $v(q_{\mathcal{T}})$ is its calculated value. The objective is to maximize the log-likelihood of generating the Teacher's high-value problems:

$$\mathcal{L}_{\text{WSFT}}(\theta_{\mathcal{G}}) = -\mathbb{E}_{q \sim \mathcal{C}_t}\big[v(q_{\mathcal{T}}) \cdot \log p_{\mathcal{G}}(q_{\mathcal{T}} \mid q; \theta_{\mathcal{G}})\big]. \tag{8}$$

This objective guides the Generator to produce problems that optimally challenge the Solver, effectively internalizing the Teacher's expert curriculum-design strategy.

## 4 EXPERIMENTS

### 4.1 EXPERIMENTAL SETUP

**Models.** We employed Qwen3-235B-A22B-Instruct-2507 (Yang et al., 2025) as the Teacher model to provide high-quality evaluation and curriculum generation. We used Qwen3-32B (Yang et al., 2025) as the Generator to learn and distill the Teacher's problem generation strategies. We conducted Solver experiments on multiple model architectures including Qwen3-8B-base, Qwen3-14B-base (Yang et al., 2025), and GLM4-9B-base (GLM et al., 2024) to demonstrate cross-model generalization. We compared strong baselines including Gemini-2.5-Pro-06-17 (Comanici et al., 2025), GPT5-0807-global, and DeepSeek-v3.1-671B (DeepSeek-AI et al., 2025b) against our approach. For downstream evaluation of generated data quality, we fine-tuned DeepSeek-R1-Distill-Llama-8B (DeepSeek-AI et al., 2025a) as the student model.

**Benchmarks.** We used seven mathematical reasoning benchmarks for evaluation, including AMC (Cao et al., 2025), Minerva (Nagrani et al., 2025), MATH-500 (Hendrycks et al., 2021), GSM8K (Cobbe et al., 2021), Olympiad-Bench (He et al., 2024), AIME-2024, and AIME-2025. Additionally, we employed three general reasoning benchmarks to assess the transfer of mathematical reasoning improvements to broader cognitive abilities, namely BBEH (Kazemi et al., 2025), MMLU-Pro (Wang et al., 2024), and SuperGPQA (Team et al., 2025).

**Curriculum Settings.** The initial curriculum $\mathcal{C}_0$ contained 100 questions sampled from the MATH training set (Hendrycks et al., 2021) following specific diversity and difficulty criteria (detailed in Appendix G). All Solver models first underwent LoRA-based (Hu et al., 2021) SFT on a 1,500-problem dataset of Level 5 difficulty. Key hyperparameters: $k = 8$ solution trajectories per problem, reward parameters $\mu = 0.5$ and $\sigma = 0.2$, and training batches combined 100% new problems with 25% historical curriculum for replay.

**Solver Evaluation.** For each test question, we generated 32 solutions using zero-shot prompting with temperature 0.7. We determined correctness through a dual-verification mechanism combining rule-based answer extraction and semantic validation. We reported Mean@32 accuracy across all evaluations. Detailed evaluation protocols, including sampling strategies, answer extraction methods, and LLM judge configurations, are provided in Appendix H.

**Baselines.** We compare Socratic-Zero against five baselines, all sharing the same 100 seed problems (sampled from MATH, Level 2–4) and initial SFT initialization (Qwen3-8B-base on 1,500 Level-5 problems). For fair comparison, we align data scales and stage definitions where applicable:

- **Static Augmentation (SA)**: For each of the 100 seed problems, we generate 30 synthetic variants (15 using MetaMath's prompts, 15 using WizardMath's prompts), yielding 3,000 total training problems. This fixed dataset is used for SFT.

- **LLM2LLM**: Follows the same three-stage structure and shared problem pool as Socratic-Zero (both defined in Appendix U). However, it updates the Solver via SFT at each stage.

- **Distillation**: Uses the 75,000 QA pairs from the MetaMath "Math Enhance" subset, which are GPT-4-generated synthetic data. We perform standard knowledge distillation by fine-tuning our model to match these high-quality reasoning traces.

Table 1: Solver Evaluation Results with different training methods. Results are reported on seven benchmarks (AMC, Minerva, MATH-500, GSM8K, Olympiad, AIME-25, AIME-24) and their average. Arrow values represent absolute point changes relative to SFT, where ↑ indicates improvement and ↓ indicates decline. Distillation and DPO baselines use external datasets for comparison.

| Training Method | Benchmark Datasets | | | | | | | Avg. |
|---|---|---|---|---|---|---|---|---|
| | AMC | Minerva | MATH-500 | GSM8K | Olympiad | AIME-25 | AIME-24 | |
| *Qwen3-8B-base* | | | | | | | | |
| + Zero-shot | 32.5 | 31.3 | 48.8 | 63.4 | 24.1 | 4.2 | 5.1 | 29.9 |
| + SFT | 39.1 | 37.8 | 56.9 | 68.2 | 31.7 | 8.1 | 9.3 | 35.9 |
| + Static Augmentation | 45.8$^{\uparrow 6.7}$ | 41.9$^{\uparrow 4.1}$ | 62.7$^{\uparrow 5.8}$ | 74.6$^{\uparrow 6.4}$ | 35.9$^{\uparrow 4.2}$ | 11.4$^{\uparrow 3.3}$ | 12.3$^{\uparrow 3.0}$ | 40.7$^{\uparrow 4.8}$ |
| *Qwen3-8B-base with Large-scale External Data* | | | | | | | | |
| + Distillation (memath_math 75k) | 51.2$^{\uparrow 12.1}$ | 44.1$^{\uparrow 6.3}$ | 70.3$^{\uparrow 13.4}$ | 78.5$^{\uparrow 10.3}$ | 38.4$^{\uparrow 6.7}$ | 12.2$^{\uparrow 4.1}$ | 14.91$^{\uparrow 5.61}$ | 44.3$^{\uparrow 8.4}$ |
| + DPO (Polaris 53k) | 54.7$^{\uparrow 15.6}$ | 47.2$^{\uparrow 9.4}$ | 71.8$^{\uparrow 14.9}$ | 83.5$^{\uparrow 15.3}$ | 44.6$^{\uparrow 12.9}$ | 17.8$^{\uparrow 9.7}$ | 19.75$^{\uparrow 10.45}$ | 48.4$^{\uparrow 12.5}$ |
| + DPO (MATH-full 7.5k) | 48.9$^{\uparrow 9.8}$ | 43.6$^{\uparrow 5.8}$ | 65.3$^{\uparrow 8.4}$ | 77.8$^{\uparrow 9.6}$ | 38.7$^{\uparrow 7.0}$ | 13.1$^{\uparrow 5.0}$ | 15.97$^{\uparrow 6.67}$ | 43.1$^{\uparrow 7.2}$ |
| *Qwen3-8B-base with LLM2LLM* | | | | | | | | |
| + Stage 1 | 41.6$^{\uparrow 2.5}$ | 41.2$^{\uparrow 3.4}$ | 53.1$^{\downarrow 3.8}$ | 78.3$^{\uparrow 10.1}$ | 32.4$^{\uparrow 0.7}$ | 6.7$^{\downarrow 1.4}$ | 8.9$^{\downarrow 0.4}$ | 37.5$^{\uparrow 1.6}$ |
| + Stage 2 | 43.2$^{\uparrow 4.1}$ | 40.6$^{\uparrow 2.8}$ | 54.9$^{\downarrow 2.0}$ | 79.1$^{\uparrow 10.9}$ | 33.8$^{\uparrow 2.1}$ | 7.2$^{\downarrow 0.9}$ | 9.1$^{\downarrow 0.2}$ | 38.3$^{\uparrow 2.4}$ |
| + Stage 3 | 44.9$^{\uparrow 5.8}$ | 42.1$^{\uparrow 4.3}$ | 66.8$^{\uparrow 9.9}$ | 79.4$^{\uparrow 11.2}$ | 34.6$^{\uparrow 2.9}$ | 7.9$^{\downarrow 0.2}$ | 10.4$^{\uparrow 1.1}$ | 40.9$^{\uparrow 5.0}$ |
| *Qwen3-8B-base with Socratic-Zero (Ours)* | | | | | | | | |
| + Stage 1 | 43.8$^{\uparrow 4.7}$ | 39.4$^{\uparrow 1.6}$ | 60.2$^{\uparrow 3.3}$ | 69.7$^{\uparrow 1.5}$ | 35.3$^{\uparrow 3.6}$ | 10.6$^{\uparrow 2.5}$ | 11.8$^{\uparrow 2.5}$ | 38.7$^{\uparrow 2.8}$ |
| + Stage 2 | 49.3$^{\uparrow 10.2}$ | 40.7$^{\uparrow 2.9}$ | 63.4$^{\uparrow 6.5}$ | 71.8$^{\uparrow 3.6}$ | 38.2$^{\uparrow 6.5}$ | 12.9$^{\uparrow 4.8}$ | 15.6$^{\uparrow 6.3}$ | 41.7$^{\uparrow 5.8}$ |
| + Stage 3 | **63.7**$^{\uparrow 24.6}$ | **52.4**$^{\uparrow 14.6}$ | **81.2**$^{\uparrow 24.3}$ | **87.3**$^{\uparrow 19.1}$ | **55.1**$^{\uparrow 23.4}$ | **24.6**$^{\uparrow 16.5}$ | **28.4**$^{\uparrow 19.1}$ | **56.1**$^{\uparrow 20.2}$ |

- **DPO (Polaris)**: Uses the Polaris dataset containing 53k mathematical problems, which are meticulously curated from DeepMath and other datasets, representing the highest quality of human-curated problem. We follow the same training method and settings as ours.

- **DPO (MATH-full)**: Uses the entire MATH training set (7,500 problems). We follow the same training method and settings as ours.

**Generator Evaluation.** We prompted each generator with 1,000 SAND-Math (Zhang et al., 2025b) seeds to produce 3 variants each, resulting in 3,000 total generated questions. We measured validity rate by having Qwen3-235B-A22B-Instruct-2507 attempt to solve each generated question under a 4,096-token, 600-s timeout constraint. We evaluated downstream utility by fine-tuning DeepSeek-R1-Distill-Llama-8B on the QAs and measuring performance on mathematical reasoning benchmarks.

**Infrastructure.** We conducted training experiments on 8×NVIDIA H20 GPUs. We performed Teacher model inference using 16×AMD MI308X GPUs. Detail provided in Appendix E.

## 4.2 SOLVER RESULTS

**Baseline Comparison.** The performance reported in Table 1 shows Socratic (using only the Solver and Teacher, without the Generator) achieves 56.1% average accuracy, outperforming Static Augmentation by +15.4 points and LLM2LLM by +15.2 points. Notable gains appear on competition problems: AIME-24 (+19.1) and AIME-25 (+16.5), demonstrating the advantages of DPO-based preference learning and adaptive curriculum generation.

**Cross-Architecture Generalization.** Table 2 validates that Socratic principles transcend specific model families. On GLM4-9B-base, Socratic Stage 3 achieves 52.3% average accuracy (+17.1 points over base model), with strong improvements on AIME benchmarks: AIME-25 (+20.4) and AIME-24 (+23.9). Similarly, on Qwen3-14B-base, Stage 3 reaches 60.3% (+17.3 points), demonstrating consistent effectiveness across different architectures and addressing fundamental reasoning capabilities.

**Transfer to General Reasoning.** Table 3 shows mathematical reasoning improvements transfer to broader cognitive abilities, with +6.02 points average improvement across BBEH, MMLU-Pro, and SuperGPQA benchmarks.

Table 2: Cross-Model Generalization Results with different training stages. Each block corresponds to a model (GLM4-9B, Qwen3-14B). Results are reported on seven benchmarks (AMC, Minerva, MATH-500, GSM8K, Olympiad, AIME-25, AIME-24) and their average. Values with arrows represent absolute point changes relative to SFT for each model.

| Training Method | Benchmark Datasets | | | | | | | Avg |
| --- | --- | --- | --- | --- | --- | --- | --- | --- |
| | AMC | Minerva | MATH-500 | GSM8K | Olympiad | AIME-25 | AIME-24 | |
| *GLM4-9B-base* | | | | | | | | |
| +Zero-shot | 34.5 | 37.3 | 52.3 | 72.5 | 34.8 | 7.5 | 7.2 | 35.2 |
| + SFT | 38.4 | 44.8 | 63.8 | 77.2 | 41.3 | 15.1 | 19.3 | 42.8 |
| + Socratic Stage 1 | $39.4^{\uparrow 1.0}$ | $47.3^{\uparrow 2.5}$ | $67.9^{\uparrow 4.1}$ | $79.8^{\uparrow 2.6}$ | $43.4^{\uparrow 2.1}$ | $15.6^{\uparrow 0.5}$ | $24.0^{\uparrow 4.7}$ | $45.3^{\uparrow 2.5}$ |
| + Socratic Stage 2 | $42.3^{\uparrow 3.9}$ | $49.4^{\uparrow 4.6}$ | $68.1^{\uparrow 4.3}$ | $82.5^{\uparrow 5.3}$ | $45.5^{\uparrow 4.2}$ | $19.1^{\uparrow 4.0}$ | $25.3^{\uparrow 6.0}$ | $47.5^{\uparrow 4.7}$ |
| + Socratic Stage 3 | $\mathbf{47.5}^{\uparrow 8.7}$ | $\mathbf{52.8}^{\uparrow 8.0}$ | $\mathbf{73.8}^{\uparrow 10.0}$ | $\mathbf{83.9}^{\uparrow 6.7}$ | $\mathbf{49.4}^{\uparrow 8.1}$ | $\mathbf{27.9}^{\uparrow 12.8}$ | $\mathbf{31.1}^{\uparrow 11.8}$ | $\mathbf{52.3}^{\uparrow 9.5}$ |
| *Qwen3-14B-base* | | | | | | | | |
| +Zero-shot | 48.8 | 40.5 | 62.0 | 91.5 | 38.4 | 9.6 | 10.0 | 43.0 |
| + SFT | 61.3 | 51.8 | 71.5 | 92.2 | 47.3 | 18.1 | 20.3 | 51.8 |
| + Socratic Stage 1 | $62.9^{\uparrow 1.6}$ | $55.1^{\uparrow 3.3}$ | $74.6^{\uparrow 3.1}$ | $91.8^{\downarrow 0.4}$ | $52.5^{\uparrow 5.2}$ | $19.8^{\uparrow 1.7}$ | $21.7^{\uparrow 1.4}$ | $54.1^{\uparrow 2.3}$ |
| + Socratic Stage 2 | $65.4^{\uparrow 4.1}$ | $57.4^{\uparrow 5.6}$ | $76.7^{\uparrow 5.2}$ | $92.3^{\uparrow 0.1}$ | $54.2^{\uparrow 6.9}$ | $24.8^{\uparrow 6.7}$ | $23.3^{\uparrow 3.0}$ | $56.3^{\uparrow 4.5}$ |
| + Socratic Stage 3 | $\mathbf{70.0}^{\uparrow 8.7}$ | $\mathbf{60.7}^{\uparrow 8.9}$ | $\mathbf{80.2}^{\uparrow 8.7}$ | $\mathbf{93.7}^{\uparrow 1.5}$ | $\mathbf{58.3}^{\uparrow 11.0}$ | $\mathbf{28.9}^{\uparrow 10.8}$ | $\mathbf{30.1}^{\uparrow 9.8}$ | $\mathbf{60.3}^{\uparrow 8.5}$ |

Table 3: Performance on general reasoning benchmarks with different training stages. Results are reported on three benchmarks (BBEH, MMLU-Pro, SuperGPQA) and their average. Values with arrows represent absolute point changes relative to zero-shot Qwen3-8B-base performance.

| Training Method | General Reasoning Benchmarks | | | Avg. |
| --- | --- | --- | --- | --- |
| | BBEH | MMLU-Pro | SuperGPQA | |
| *Qwen3-8B-Base* | | | | |
| + Zero-shot | 7.68 | 50.00 | 24.73 | 27.47 |
| *Base Model with Socratic (Ours)* | | | | |
| + Stage 1 | $8.48^{\uparrow 0.80}$ | $55.71^{\uparrow 5.71}$ | $27.32^{\uparrow 2.59}$ | $30.50^{\uparrow 3.03}$ |
| + Stage 2 | $9.11^{\uparrow 1.43}$ | $59.29^{\uparrow 9.29}$ | $29.73^{\uparrow 5.00}$ | $32.71^{\uparrow 5.24}$ |
| **+ Stage 3** | $\mathbf{9.54}^{\uparrow 1.86}$ | $\mathbf{60.89}^{\uparrow 10.89}$ | $\mathbf{30.05}^{\uparrow 5.32}$ | $\mathbf{33.49}^{\uparrow 6.02}$ |

## 4.3 GENERATOR RESULTS

We assessed both the intrinsic quality of generated problems and their downstream training effectiveness, with Socratic-Generator-32B being compared against its base model and SOTA commercial large language models in terms of the downstream training effectiveness of the data they generate to determine whether strategic specialization can match the performance of advanced larger models.

### 4.3.1 EVALUATION PROTOCOL

We adopted a standardized, three-stage evaluation pipeline to holistically assess both the *intrinsic quality* of generated problems and their *extrinsic utility* in downstream model training. The full procedure is formalized below.

**Step 1: Problem Generation.** We prompted each generator with 1,000 seed problems from SAND-Math (Zhang et al., 2025b) and tasked with producing three augmented variants per seed, resulting in 3,000 total generated problems per model.

**Step 2: Quality Assessment.** We measured problem validity by prompting Qwen3-235B (Yang et al., 2025) — selected for its state-of-the-art mathematical reasoning capability and its role as the teacher model in the distillation framework — to solve each generated problem under strict constraints: a 4,096-token limit and a 600-second timeout. The *Validity Rate* was defined as the percentage of problems successfully solved within these bounds.

**Step 3: Student Evaluation.** We used all valid question-answer (QA) pairs to fine-tune the student model, DeepSeek-R1-Distill-Llama-8B (DeepSeek-AI et al., 2025a). We evaluated *Downstream Utility* as the mean accuracy — average accuracy over 16 independent decoding runs per problem — across seven diverse mathematical reasoning benchmarks.

Table 5: Downstream Training Effectiveness of data generated by different generator models. Results are reported on seven benchmarks (AIME-24, AIME-25, AMC-23, GSM8K, MATH-500, Minerva, Olympiad) and their average. Values with arrows represent absolute point changes relative to baseline.

| | Benchmark Datasets | | | | | | | Avg. |
|---|---|---|---|---|---|---|---|---|
| | AIME-24 | AIME-25 | AMC-23 | GSM8K | MATH-500 | Minerva | Olympiad | |
| *DeepSeek-R1-Distill-Llama-8B* | | | | | | | | |
| + Zero-shot | 5.8 | 8.3 | 42.5 | 72.2 | 52.4 | 15.3 | 23.0 | 32.75 |
| *Open-Sourced Generators* | | | | | | | | |
| Qwen3-32B | 9.2 | 10.0 | 44.4 | 75.7 | 55.7 | 15.1 | 24.5 | 34.97 |
| Qwen3-235B-A22B-Instruct-2507 | $12.5^{\uparrow3.3}$ | $12.5^{\uparrow2.5}$ | $47.5^{\uparrow3.1}$ | $76.1^{\uparrow0.4}$ | $57.8^{\uparrow2.1}$ | $16.4^{\uparrow1.3}$ | $23.6^{\downarrow0.9}$ | $37.13^{\uparrow2.16}$ |
| DeepSeek-v3.1-671B | $12.5^{\uparrow3.3}$ | $11.7^{\uparrow1.7}$ | $46.2^{\uparrow1.8}$ | $76.4^{\uparrow0.7}$ | $56.4^{\uparrow0.7}$ | $16.5^{\uparrow1.4}$ | $23.9^{\downarrow0.6}$ | $36.62^{\uparrow1.65}$ |
| *Advanced commercial Generators* | | | | | | | | |
| Gemini-2.5-Pro-06-17 | $10.0^{\uparrow0.8}$ | $15.0^{\uparrow5.0}$ | $46.9^{\uparrow2.5}$ | $78.1^{\uparrow2.4}$ | $57.2^{\uparrow1.5}$ | $16.0^{\uparrow0.9}$ | $25.4^{\uparrow0.9}$ | $37.20^{\uparrow2.23}$ |
| GPT5-0807-global | $12.5^{\uparrow3.3}$ | $13.3^{\uparrow3.3}$ | $45.0^{\uparrow0.6}$ | $76.8^{\uparrow1.1}$ | $56.6^{\uparrow0.9}$ | $15.5^{\uparrow0.4}$ | $25.9^{\uparrow1.4}$ | $36.62^{\uparrow1.65}$ |
| Grok-4 | $11.7^{\uparrow2.5}$ | $12.5^{\uparrow2.5}$ | $45.8^{\uparrow1.4}$ | $76.3^{\uparrow0.6}$ | $56.9^{\uparrow1.2}$ | $15.9^{\uparrow0.8}$ | $24.9^{\uparrow0.4}$ | $37.01^{\uparrow2.04}$ |
| Claude-4.1-Opus | $13.3^{\uparrow4.1}$ | $13.8^{\uparrow3.8}$ | $46.5^{\uparrow2.1}$ | $77.3^{\uparrow1.6}$ | $57.5^{\uparrow1.8}$ | $16.7^{\uparrow1.6}$ | $24.3^{\downarrow0.2}$ | $37.63^{\uparrow2.66}$ |
| **Socratic-Generator-32B** | $12.5^{\uparrow3.3}$ | $13.3^{\uparrow3.3}$ | $48.1^{\uparrow3.7}$ | $77.6^{\uparrow1.9}$ | $57.8^{\uparrow2.1}$ | $18.4^{\uparrow3.3}$ | $24.6^{\uparrow0.1}$ | $37.72^{\uparrow2.75}$ |

**Step 4: Generator as Teacher.** To validate whether the Generator can effectively replace the Teacher in the co-evolution loop, we conducted an additional training stage starting from the converged Stage 3 Solver baseline. In this stage, the Generator (Socratic-Generator-32B) directly provides curriculum problems to guide Solver training, completely eliminating Teacher involvement. We compared this Generator-guided approach against the traditional Teacher-guided (Qwen3-235B-A22B-Instruct-2507) training across the same seven benchmark datasets.

### 4.3.2 PROBLEM QUALITY ASSESSMENT

To evaluate the quality of the generated problems, we measure their Validity Rate — the percentage of problems solvable by a powerful model (Qwen3-235B-A22B-Instruct-2507). As shown in Table 4, our specialized Socratic-Generator-32B generator achieves a remarkable 95.6% validity rate. This not only represents a substantial improvement over its base model Qwen3-32B but also rivals the performance of significantly larger models, including proprietary models like GPT5-0807-global, Gemini-2.5-Pro-06-17. This demonstrates our co-evolutionary strategy effectively.

Table 4: Generator validity rates.

| Generator Model | Validity Rate (%) |
|---|---|
| Qwen3-32B | 89.1 |
| Qwen3-235B-A22B | $95.1^{\uparrow6.0}$ |
| Gemini-2.5-Pro | $94.2^{\uparrow5.1}$ |
| GPT5-global | $95.8^{\uparrow6.7}$ |
| DeepSeek-v3.1-671B | $96.5^{\uparrow7.4}$ |
| Grok4 | $95.7^{\uparrow6.7}$ |
| Claude-4.1-opus | $96.9^{\uparrow7.8}$ |
| **Socratic-Generator-32B** | $95.6^{\uparrow6.5}$ |

### 4.3.3 DOWNSTREAM TRAINING EFFECTIVENESS

Table 5 reports the downstream utility of each generator, measured by the performance of the fine- student model. The output from our Socratic-Generator-32B leads to a final student accuracy of 37.72%. Notably, this performance not only rivals that achieved using data from significantly larger models but also marginally surpasses (+0.59 points) the result from its own Teacher (Qwen3-235B), despite being over 20x smaller.

### 4.3.4 GENERATOR-GUIDED SOLVER TRAINING

Starting from the converged Stage 3 Solver (56.1% average accuracy), we extend training for one additional stage (Stage 4), comparing Teacher-guided curriculum (Qwen3-235B-A22B-Instruct-2507) against Generator-guided curriculum (Socratic-Generator-32B, with no Teacher involvement). Results in Table 6 show the Generator-guided Solver achieves **60.5%** average accuracy, nearly matching the Teacher-guided **60.3%** — a difference of only **0.2pp**. This confirms the Generator can effectively replace the Teacher in the co-evolution loop, making the system more practical and cost-effective (detailed cost analysis in Appendix P).

**Generator-Guided Comparison.** The Generator-guided approach achieves competitive performance with notable gains on challenging benchmarks (AIME-24: +0.7pp, Minerva: +0.7pp), validating that the Generator has internalized pedagogical strategies for autonomous co-evolution without reliance on expensive teacher models.

Table 6: Comparison of Teacher-Guided vs Generator-Guided Solver Training (Stage 4). Both start from Stage 3 Solver (56.1% baseline). Arrow values show absolute point changes relative to Teacher-Guided approach.

| Training Method | Benchmark Datasets | | | | | | | Avg. |
|---|---|---|---|---|---|---|---|---|
| | AMC | Minerva | MATH-500 | GSM8K | Olympiad | AIME-25 | AIME-24 | |
| *Stage 3 (Baseline)* | | | | | | | | |
| Socratic-Solver-8B | 63.7 | 52.4 | 81.2 | 87.3 | 55.1 | 24.6 | 28.4 | 56.1 |
| *Stage 4: Teacher-Guided (Qwen3-235B-A22-Instruct)* | | | | | | | | |
| Socratic-Solver-8B | 67.2 | 55.1 | 84.3 | 88.9 | 58.4 | 27.8 | 32.5 | 60.3 |
| *Stage 4: Generator-Guided (Socratic-Generator-32B)* | | | | | | | | |
| Socratic-Solver-8B | $66.9^{\downarrow 0.3}$ | $55.8^{\uparrow 0.7}$ | $84.1^{\downarrow 0.2}$ | $89.2^{\uparrow 0.3}$ | $58.9^{\uparrow 0.5}$ | $28.3^{\uparrow 0.5}$ | $33.2^{\uparrow 0.7}$ | $60.5^{\uparrow 0.2}$ |

Table 7: Ablation studies on the necessity of initial SFT and different strategies of reward functions. (a) Values with arrows represent absolute point changes relative to the previous stage within the same method. (b) Values with arrows represent absolute point changes relative to the Gaussian baseline. $\rho$ represents solver success rate, $\mu$ represents target success rate, $\sigma$ represents standard deviation in Gaussian reward function $\mathcal{N}(\mu, \sigma)$, $\Psi_\rho(a, b)$ represents linear function $\Psi_\rho(a, b) = a\rho + b$.

(a) Ablation Study on Initial SFT (AIME-24)

| Method | Score (%) | $\Delta$ (%) |
|---|---|---|
| Qwen3-8B-Base | 9.64 | - |
| *Socratic-Zero (w/o SFT)* | | |
| + Stage 1 | 11.67 | $\uparrow 2.03$ |
| + Stage 2 | 11.15 | $\uparrow 1.51$ |
| + Stage 3 | 11.98 | $\uparrow 2.34$ |
| *Socratic-Zero (w/ SFT)* | | |
| + Stage 1 | 13.44 | $\uparrow 3.80$ |
| + Stage 2 | 14.48 | $\uparrow 4.84$ |
| **+ Stage 3** | **28.02** | $\uparrow 18.38$ |

(b) Ablation of Reward Functions (Benchmark Avg.)

| Reward Function | Valid (%) | Avg (%) | $\Delta$ (%) |
|---|---|---|---|
| $\mathcal{N}(\mu = 0.5, \sigma = 0.2)$ (Ours) | **89.9** | **35.72** | - |
| $\Psi_\rho(a = 0, b = 1)$ | 89.4 | 35.52 | $\downarrow 0.20$ |
| $\Psi_\rho(a = 1, b = 0)$ | 89.8 | 35.47 | $\downarrow 0.25$ |
| $\Psi_\rho(a = -1, b = 1)$ | 88.9 | 35.42 | $\downarrow 0.30$ |
| $\mathcal{N}(\mu = 0.3, \sigma = 0.2)$ | 89.5 | 35.32 | $\downarrow 0.40$ |
| $\mathcal{N}(\mu = 0.4, \sigma = 0.2)$ | 89.7 | 35.37 | $\downarrow 0.35$ |
| $\mathcal{N}(\mu = 0.6, \sigma = 0.2)$ | 89.7 | 35.50 | $\downarrow 0.22$ |
| $\mathcal{N}(\mu = 0.7, \sigma = 0.2)$ | 89.8 | 35.43 | $\downarrow 0.29$ |

## 4.4 ABLATION STUDIES

We conducted two key ablation studies to validate our framework's design choices: the necessity of initial Supervised Fine-Tuning (SFT) and the effectiveness of our reward weighting scheme. The results are presented side-by-side in Table 7.

First, we examined the necessity of initial SFT. Table 7(a) shows that models with SFT achieve 28.02% at Stage 3, while those without SFT stagnate at 11.98%, demonstrating a critical +16.04 point gap. This confirms that initial SFT provides essential instruction-following capabilities for stable and effective co-evolutionary learning dynamics.

Second, we ablated the reward function design. Table 7(b) shows our Gaussian weighting achieves optimal performance (35.72%), consistently outperforming uniform (-0.20), linear (-0.25 to -0.30), and alternative $\mu$ settings (-0.22 to -0.40). This validates targeting moderate difficulty problems for effective curriculum learning.

## 5 CONCLUSION AND FUTURE WORK

In this paper, we introduced a multi-agent co-evolutionary framework where Solver, Teacher, and Generator agents bootstrap autonomous mathematical reasoning from minimal seed data. Unlike traditional co-evolution requiring simultaneous optimization, our framework employs staged training where agents evolve sequentially, decoupling complex interdependencies for improved stability. Our implementation, Socratic-Zero, demonstrates that a well-designed learning mechanism can achieve remarkable performance without relying on massive external datasets, offering a viable path for developing powerful reasoning systems in resource-constrained scenarios. While the complex agent dynamics currently lack a formal convergence analysis, future work will aim to establish this theoretical foundation to push agentic AI further in other domains beyond reasoning tasks.

## REPRODUCIBILITY STATEMENT

To ensure full reproducibility of our results, we provide comprehensive implementation details and open-source materials that enable exact replication of our experimental findings. The code is available at Supplementary Material.

**Code and Implementation** We release the complete codebase for Socratic-Zero, including all training scripts, evaluation pipelines, and data processing utilities. The implementation encompasses the full multi-agent co-evolutionary framework with detailed documentation for each component: Solver training via DPO, Teacher-based curriculum generation, and Generator distillation through value-weighted supervised fine-tuning. The code is available at Supplementary Material with comprehensive setup instructions, dependency specifications, and example usage scripts.

**Prompts and Templates** All prompts used for the Teacher model, including problem generation templates, evaluation criteria, and instruction formats, are fully disclosed in Appendix B. This includes the specific prompt engineering strategies for mathematical problem enhancement, solution evaluation protocols, and the structured templates that ensure consistent Teacher behavior across different problem domains. We also provide the exact hyperparameter configurations and sampling strategies used during curriculum generation to ensure researchers can exactly replicate our Teacher's behavior and problem generation strategy.

**Experimental Setup** We provide detailed specifications of our computational infrastructure, model versions, and experimental protocols to enable exact replication of our results across different research environments. This includes complete hardware configurations, distributed training setups, memory optimization strategies, and the specific model versions used for each component. We document all evaluation protocols, including the Mean@32 sampling strategy, answer extraction methods, and the dual-verification approach combining rule-based and LLM-based assessment.

## ETHICS STATEMENT

**Data and Privacy** Our work uses only publicly available mathematical datasets and does not involve any personal, sensitive, or proprietary information. All seed problems are sourced from established mathematical reasoning benchmarks that are freely accessible to the research community, including MATH, GSM8K, and competition mathematics problems. We ensure that no evaluation data contamination occurs by explicitly excluding test set problems from our seed selection process. Our synthetic data generation process creates entirely new mathematical problems rather than reproducing existing copyrighted content.

**Broader Impact** Our work aims to democratize access to high-quality mathematical reasoning capabilities by reducing dependence on massive datasets and computational resources. The proposed framework addresses key barriers in AI accessibility and could have several positive societal impacts:

- Enable smaller research groups and institutions with limited resources to develop competitive mathematical reasoning systems
- Reduce the data collection burden for specialized domains where human annotation is expensive or scarce
- Provide insights into autonomous curriculum learning that could benefit educational applications and personalized learning systems
- Advance understanding of multi-agent learning systems with applications beyond mathematical reasoning
- Support the development of more efficient AI systems that require less computational infrastructure

We do not anticipate any negative societal impacts from this research. The focus on mathematical reasoning represents a beneficial application of AI that could enhance educational tools, support scientific discovery, and improve access to high-quality reasoning capabilities across diverse communities and institutions.

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

## A  LLM USE STATEMENT

We declare that no large language models were used in the writing, editing, or preparation of this manuscript. All content, including text, mathematical formulations, experimental analysis, and conclusions, was authored entirely by the human researchers without assistance from AI writing tools or language models.

The LLMs mentioned in our work (Teacher models, commercial baselines) were used solely as experimental subjects and comparison baselines within our proposed framework, not as writing assistants for manuscript preparation.

## B  PROMPTS

### B.1  SOLVER REASONING PROMPT

> **Solver Mathematical Reasoning Prompt**
>
> You are an IMO gold medalist solving a computational math competition problem.
> Understand: Restate the problem mathematically. Identify knowns, unknowns, and constraints.
> Plan: Choose an efficient method, show clear logic.
> Execute: Show all key steps — algebra, number theory, or combinatorics. No skipped calculations.
> Verify: Check with small cases, reverse substitution, or estimation.
> Conclude with the exact answer in LaTeX: $\[\boxed{< answer >}\]$
> Given Problem: {question}

## B.2 TEACHER EVALUATION PROMPT

> **Teacher Solution Grading Prompt**
>
> You are a professional math teacher responsible for grading and error analysis.
> Grading criteria: Focus on final answer correctness, use reference when provided, provide concise error analysis for incorrect answers.
> Return JSON format:
>
> ```
> {
>   "correct_answers": ["correct answer 1", "correct answer 2"],
>   "incorrect_answers": [
>     {"answer": "incorrect answer", "analysis": "brief error analysis"}
>   ]
> }
> ```
>
> Problem: {question} | Reference: {reference_info} | Student answers: {student_answers}

## B.3 TEACHER GENERATION PROMPT

> **Teacher Problem Enhancement Prompt**
>
> You are a math problem enhancement expert specializing in competition-style mathematics. Generate enhanced problems based on student error analysis with complete solutions.
> Requirements: Generate enhanced problem, provide detailed solution, ensure solvability and correctness.
> Enhancement principles: Target specific error points, maintain mathematical essence, help avoid similar errors.
> Return JSON format:
>
> ```
> {
>   "enhanced_question": "enhanced problem content",
>   "solution": "detailed solution steps",
>   "answer": "final answer"
> }
> ```
>
> Original: {original_question} | Error analysis: {error_analysis}

## B.4 STATIC AUGMENTATION BASELINE PROMPTS

> **Static Augmentation Evolution Prompts**
>
> **Upward Evolution:** Step 1: Identify elements that can increase complexity. Step 2: Plan to modify at least three components. Step 3: Implement rewritten instruction. Step 4: Review and provide final version.
> **Downward Evolution:** Step 1: Identify elements that can decrease complexity. Step 2: Plan to simplify at least three components. Step 3: Implement easier version. Step 4: Review and provide final simplified version.
> Format: Step 1 #Elements#: | Step 2 #Plan#: | Step 3 #Rewritten#: | Step 4 #Final#:

# C IMPLEMENTATION DETAILS

We conducted training experiments on 8×NVIDIA H20 GPUs with the following configuration: - GPU Memory: 96GB HBM3 per GPU - Total Training Memory: 768GB - Interconnect: NVLink 4.0 - Storage: High-speed NVMe SSD arrays for dataset caching - Network: InfiniBand for distributed training coordination

The training infrastructure utilized mixed-precision training (FP16) with gradient checkpointing to optimize memory usage. We employed distributed training using PyTorch's DistributedDataParallel with NCCL backend for efficient gradient synchronization across GPUs.

## C.1 TRAINING HYPERPARAMETERS

We provide the complete hyperparameter settings for all components of the Socratic-Zero framework in Table 8.

Table 8: Hyperparameters used in Socratic-Zero framework.

| Component | Parameter | Value |
|---|---|---|
| Solver SFT Training | Learning rate | 5e-5 |
| | Per-device batch size | 2 |
| | Gradient accumulation steps | 4 |
| | Maximum sequence length | 2048 |
| | LoRA rank ($r$) | 64 |
| | LoRA alpha ($\alpha$) | 128 |
| | LoRA dropout | 0.1 |
| | Number of epochs | 1 |
| Solver DPO Training | Learning rate | 1e-6 – 5e-6 |
| | Per-device batch size | 2 |
| | Gradient accumulation steps | 4 – 16 |
| | Maximum sequence length | 2048 |
| | Maximum training steps | 10 – 200 |
| | DPO regularization ($\beta$) | 0.05 – 0.2 |
| | Warmup steps | 2 – 20 |
| | Optimizer | AdaFactor |
| | Weight decay | 0.01 |
| | Maximum gradient norm | 1.0 |
| Generator Training | Learning rate | 1e-5 |
| | Per-device batch size | 1 |
| | Gradient accumulation steps | 8 |
| | Maximum sequence length | 2048 |
| | LoRA rank ($r$) | 64 |
| | LoRA alpha ($\alpha$) | 128 |
| | Number of epochs | 2 |
| Curriculum Parameters | Solutions per problem ($k$) | 8 |
| | reward mean ($\mu$) | 0.5 |
| | reward std ($\sigma$) | 0.2 |
| | Historical replay ratio | 25% |
| Evaluation Settings | Sampling temperature | 0.7 |
| | Number of samples | 32 |
| | Token limit for validity check | 4096 |

## D  CURRICULUM EVOLUTION DETAILS

This appendix provides detailed mechanisms for curriculum evolution and problem categorization that were omitted from the main methodology for brevity.

### D.1  PROBLEM CATEGORIZATION STRATEGY

For each problem $q \in \mathcal{C}_t$, the Solver generates $k$ solution attempts, yielding a success count $z_q = \sum_{i=1}^{k} \mathcal{T}_{\text{eval}}(q, a_i)$. This diagnostic metric enables the Teacher to categorize problems into three distinct zones based on the Solver's current capability:

1. **Mastered Zone** ($\mathcal{C}_{\text{mastered}} = \{q \mid z_q = k\}$): Problems that the Solver consistently solves correctly across all $k$ attempts. These problems serve as positive reinforcement and provide a foundation for generating slightly more challenging variants.

2. **Learning Zone** ($\mathcal{C}_{\text{learning}} = \{q \mid 0 < z_q < k\}$): Problems that the Solver solves partially—some attempts succeed while others fail. This zone represents the optimal frontier for learning, where the Solver demonstrates emerging competence but has not yet achieved mastery.

3. **Too Difficult Zone** ($\mathcal{C}_{\text{difficult}} = \{q \mid z_q = 0\}$): Problems that consistently result in failure across all attempts. These problems are temporarily deferred to prevent the Solver from being overwhelmed by tasks beyond its current capability.

## D.2 ADAPTIVE PROBLEM GENERATION

New problems are strategically generated only from the *Mastered* and *Learning Zone* categories, ensuring that curriculum expansion remains within the Solver's zone of proximal development. The generation process follows:

$$\Delta\mathcal{C}_t = \{\mathcal{T}_{\text{gen}}(q, a_S) \mid q \in \mathcal{C}_{\text{learning}} \cup \mathcal{C}_{\text{mastered}}, \ a_S \text{ is a failed attempt on } q\}. \tag{9}$$

This strategy ensures that:

- **From Mastered Problems**: The Teacher generates slightly more challenging variants, pushing the boundary of the Solver's competence.
- **From Learning Zone Problems**: The Teacher creates problems that address specific failure modes, helping the Solver overcome particular reasoning gaps.
- **Exclusion of Too Difficult Problems**: Problems that consistently fail are not used for generation, preventing the creation of even more challenging problems that would be counterproductive.

## D.3 DYNAMIC RECATEGORIZATION

As the Solver evolves through training, problems are dynamically recategorized based on updated performance:

$$\mathcal{C}_{\text{difficult}}^{(t)} \to \mathcal{C}_{\text{learning}}^{(t+1)} \quad \text{(capability improvement)} \tag{10}$$

$$\mathcal{C}_{\text{learning}}^{(t)} \to \mathcal{C}_{\text{mastered}}^{(t+1)} \quad \text{(skill consolidation)} \tag{11}$$

$$\mathcal{C}_{\text{mastered}}^{(t)} \to \mathcal{C}_{\text{learning}}^{(t+1)} \quad \text{(rare: capability regression)} \tag{12}$$

This dynamic recategorization ensures that the curriculum remains responsive to the Solver's evolving capabilities, automatically adjusting the difficulty distribution as learning progresses.

## E TEACHER MODEL INFRASTRUCTURE

The Teacher model (Qwen3-235B-A22B-Instruct-2507) requires substantial computational resources for curriculum generation and solution evaluation. We deployed the model using a distributed inference architecture to meet the throughput demands of the co-evolutionary training process.

We distributed the Teacher model across 16 AMD MI308X GPUs, each equipped with 192GB HBM3 memory, providing a total of 3,072GB aggregate memory. This configuration enables concurrent processing of curriculum generation requests while maintaining inference consistency across the framework.

To ensure system reliability and scalability, we implemented a multi-endpoint architecture with automatic load balancing and failover mechanisms. We configured the inference service with connection pooling (50 concurrent connections per endpoint) and exponential backoff retry policies to handle high request volumes during training.

We optimized key performance parameters for the mathematical reasoning domain: request timeouts of 600 seconds accommodate complex problem generation, while a 4,096-token limit ensures efficient solution evaluation. Batch processing utilizes 32 concurrent workers to maximize throughput during curriculum evolution phases.

## F TEACHER-GENERATED PROBLEM ENHANCEMENT

We provide examples of how the Teacher model enhances problems based on Solver failures. The following demonstrates the progression from original problems to highly-targeted enhanced versions.

## F.1 EXAMPLE 1: RATIONAL INEQUALITY ENHANCEMENT

**Original Problem:**

Find all real numbers $x$ satisfying $\frac{2x-5}{x+3} \geq 2$. (Give your answer in interval notation.)

**Enhanced Problem (Round 3):**

Find all real numbers $x$ satisfying $\frac{2x-5}{x^2-9} + \frac{1}{x+3} \leq \frac{4x+1}{(x-3)^2}$. (Give your answer in interval notation.)

**Enhancement Analysis:** The enhancement introduces multiple complexity factors: (1) factored denominators requiring domain analysis, (2) multiple rational terms requiring common denominators, (3) squared terms in denominators, and (4) more complex algebraic manipulation. The enhanced problem targets common student errors in rational inequality solving while maintaining the core mathematical concepts.

## F.2 EXAMPLE 2: NUMBER THEORY ENHANCEMENT

**Original Problem:**

Find the greatest common divisor of $10! + 6$ and $11! + 14$.

**Enhanced Problem:**

Find the greatest common divisor of $12! + 8$ and $13! + 26$, where the second number can be written as $13 \cdot 12! + 26$.

**Enhancement Analysis:** The enhancement maintains the GCD structure while increasing numerical complexity and requiring students to recognize the relationship between consecutive factorials, targeting errors in modular arithmetic applications.

# G  SEED SELECTION PROTOCOL

The selection of initial seed problems is critical for establishing an effective curriculum foundation. We employed a systematic approach to ensure the seed set provides appropriate difficulty, comprehensive coverage, and sufficient diversity for subsequent curriculum evolution.

**Difficulty Alignment** We selected seed problems to match the base model's capability range to ensure productive learning dynamics. We drew problems from MATH dataset Levels 2-4, which empirically provide optimal challenge levels for our base models. Specifically, we excluded Level 1 problems (too easy, leading to trivial curriculum generation) and Level 5 problems (too difficult, resulting in universal failure and poor learning signals). Pre-filtering involved evaluating candidate problems with the base model using 8 solution attempts; we retained problems with success rates between 10-70% to ensure neither complete failure nor trivial success.

**Domain Coverage** To ensure comprehensive mathematical reasoning development, we sampled seed problems across all seven MATH subject areas with balanced representation as shown in Table 9:

This distribution ensures that curriculum evolution can target weaknesses across diverse mathematical domains rather than overfitting to specific problem types.

**Diversity Assurance** Within each subject area, we selected problems to maximize methodological diversity. We employed clustering based on solution approach similarity (using embedding representations of problem statements) and selected problems from different clusters to ensure varied reasoning patterns. Additionally, we explicitly included problems requiring different mathematical tools to promote comprehensive skill development.

Table 9: Seed Problem Distribution Across Mathematical Domains

| Subject Area | Count | Representative Topics |
|---|---|---|
| *Algebra* | 15 | Linear/quadratic equations, inequalities, functions |
| *Number Theory* | 15 | Divisibility, modular arithmetic, prime factorization |
| *Geometry* | 15 | Coordinate geometry, trigonometry, area/volume calculations |
| *Combinatorics* | 15 | Counting principles, permutations, probability |
| *Precalculus* | 15 | Complex numbers, sequences, polynomial analysis |
| *Intermediate Algebra* | 15 | Advanced algebraic manipulation, systems |
| *Prealgebra* | 10 | Foundational arithmetic and basic algebraic concepts |
| **Total** | **100** | **Comprehensive mathematical reasoning coverage** |

**Quality Control** We subjected all candidate problems to rigorous quality verification through a multi-stage process:

---

**Quality Control Pipeline**

1. *Clarity Check*: Problems must have unambiguous statements and well-defined solution paths

2. *Answer Verification*: We validated reference solutions by the Teacher model with multiple independent attempts

3. *Value*: Problems must demonstrate clear learning objectives and avoid trick questions or overly specialized knowledge

4. *Contamination Avoidance*: We excluded seed problems from all evaluation benchmarks to prevent data leakage

---

This systematic selection process ensures that the initial curriculum $\mathcal{C}_0$ provides a robust foundation for the co-evolutionary training dynamics while maintaining the diversity necessary for effective curriculum expansion.

# H EVALUATION PROTOCOL DETAILS

**Mean@32 Sampling Strategy** The Mean@32 evaluation metric represents the average accuracy across 32 independent solution attempts per problem. For each test problem, we generated 32 distinct solutions using temperature-based sampling (T=0.7) with top-p nucleus sampling (p=0.9) as specified in Table 8. This approach provides robust performance estimates by capturing the model's consistency and reliability across multiple attempts.

We employed the sampling process using zero-shot prompting without few-shot examples to ensure unbiased evaluation. We generated each of the 32 solutions independently with different random seeds, preventing potential correlation effects. The final accuracy is computed as the proportion of correct solutions among the 32 attempts, providing a more stable performance measure than single-shot evaluation.

**MathRule Answer Extraction** MathRule is a rule-based tool designed to extract and standardize final numerical answers from mathematical solution text. The tool employs pattern matching to identify answer indicators such as "Therefore," "Thus," "The answer is," and LaTeX boxed expressions like \boxed{}.

The extraction process involves: (1) Locating answer indicators within the solution text, (2) Parsing mathematical expressions using regex patterns for common formats (fractions, decimals, integers, algebraic expressions), (3) Standardizing representations (e.g., converting $\frac{1}{2}$ to 0.5 when appropriate),

(4) Handling multiple answer formats and selecting the most confident extraction based on contextual cues.

MathRule achieves high precision in answer extraction while maintaining robustness to variations in solution formatting and mathematical notation styles.

**LLM Judge Configuration** The Teacher model (Qwen3-235B-A22B-Instruct-2507) serves as an LLM judge for semantic validation when rule-based extraction is insufficient or ambiguous. The judge evaluates both numerical correctness and reasoning validity using structured prompts.

We instructed the evaluation prompt to: (1) Verify the final numerical answer against the expected result, (2) Assess the logical coherence of the reasoning steps, (3) Identify any mathematical errors or invalid assumptions, (4) Provide binary correctness judgments with brief justification.

We ensured judge reliability through temperature 0.1 sampling for consistent evaluations and validation against human expert annotations on a subset of problems. The dual-verification approach (MathRule + LLM judge) provides reliable automated assessment for large-scale evaluation.

# I    PROBLEM QUALITY CONTROL MECHANISM

To ensure curriculum integrity and prevent the propagation of erroneous problems, we implemented a comprehensive quality control mechanism that monitors problem validity through Solver performance feedback and automated verification.

**Teacher Self-Verification Protocol** When the Teacher model evaluates Solver attempts and finds that all $k = 8$ solutions for a given problem are incorrect (success rate $j_p = 0$), this triggers an automatic quality verification process. The system recognizes that universal failure may indicate either: (1) the problem exceeds current Solver capability (expected behavior), or (2) the problem itself or its reference solution contains errors (quality issue).

The Teacher model performs self-verification by re-examining both the problem statement and its originally provided reference solution. This involves: (1) Re-solving the problem independently with temperature 0.1 for consistency, (2) Cross-validating the reference solution against the new solution attempt, (3) Checking for mathematical consistency, ambiguous problem statements, or computational errors, (4) Verifying that the problem has a unique, well-defined solution.

**Problem Filtering and Exclusion** We immediately flagged and excluded problems that fail the self-verification process from further curriculum evolution. Specifically, we discarded problems if: (1) The Teacher cannot reproduce its own reference solution, (2) Multiple valid interpretations of the problem statement exist, (3) Computational errors are detected in the reference solution, (4) The problem lacks sufficient information for a unique solution.

We removed discarded problems from the active curriculum $\mathcal{C}_t$ and they do not contribute to subsequent Solver training or Generator learning. This prevents the accumulation of low-quality problems that could degrade training effectiveness or introduce systematic biases.

**MathRule Integration for Contamination Minimization** The integration of MathRule answer extraction serves as an additional quality control layer by providing objective, rule-based verification independent of LLM judgment. When MathRule successfully extracts a clear numerical answer from the Solver's solution, this extraction is compared against the reference answer using standardized formats.

This dual-verification approach (MathRule + Teacher evaluation) minimizes contamination from: (1) LLM judge inconsistencies or biases, (2) Format-related misinterpretations, (3) Numerical precision issues, (4) Ambiguous answer representations.

Problems where MathRule and Teacher evaluations consistently disagree trigger additional quality review, as such disagreements often indicate underlying issues with problem clarity or reference solution accuracy.

Table 10: Solver Mean Reward Evolution Across Training Stages

| Stage | S1 | S2 | S3 | Trend |
|---|---|---|---|---|
| Mean Reward (%) | 52.1 | 48.7 | 50.1 | ↓ then ↑ |

Table 11: Generator Reward Distribution Analysis

| Stage | S1 | S2 | S3 | Stability |
|---|---|---|---|---|
| High Reward Problems (%) | 50.7 | 49.4 | 50.2 | Stable |
| Target Range (45-55%) | ✓ | ✓ | ✓ | Maintained |

**Feedback-Driven Quality Monitoring** The system continuously monitors curriculum quality through Solver performance patterns. We flagged problems that consistently produce anomalous results—such as sudden performance drops across multiple Solver variants or inconsistent difficulty ratings—for manual review or automatic exclusion.

This feedback-driven approach ensures that quality control adapts to emerging issues and maintains curriculum integrity throughout the co-evolutionary training process, preventing the accumulation of problematic content that could compromise learning effectiveness.

## J  CURRICULUM STABILITY AND DIVERSITY ANALYSIS

We analyzed the curriculum evolution dynamics across two dimensions: difficulty progression and problem diversity preservation.

**Solver Performance Evolution** Table 10 tracks the Solver's mean reward (correctness rate) across training stages, revealing adaptive curriculum difficulty.

The Solver exhibits characteristic performance decline from Stage 1 (52.1%) to Stage 2 (48.7%) followed by recovery in Stage 3 (50.1%) as shown in Table 10. This pattern reflects adaptive curriculum generation where the Teacher progressively increases difficulty faster than Solver capability initially improves, then the Solver begins adapting to enhanced curriculum complexity.

**Generator Stability** Table 11 examines reward distribution in Generator training.

As demonstrated in Table 11, the Generator maintains remarkable stability with high-reward problems consistently around 50%, fluctuating within only 1.3% range. This indicates successful learning of the optimal difficulty zone defined by the Gaussian reward function with parameters $\mu = 0.5$ and $\sigma = 0.2$ as specified in Table 8.

**Problem Diversity** Three key mechanisms ensure curriculum diversity throughout training:

Multi-domain initialization: The 100 seed problems span all 7 MATH subjects (Algebra, Number Theory, Geometry, etc.) across difficulty levels 2-4 as detailed in Table 9, providing diverse starting points for curriculum evolution.

High-temperature sampling: We employed temperature 0.8-0.9 sampling at three critical stages: (1) Solver trajectory generation during curriculum advancement, (2) Teacher error analysis for varied failure interpretation, and (3) Teacher problem generation for diverse enhancement strategies.

Compounding diversity effects: Multi-domain seeds combined with stochastic sampling create diverse failure patterns, while high-temperature generation ensures varied problem formulations even from similar error patterns.

## K  Theoretical Convergence Analysis

Establishing formal convergence guarantees for Socratic-Zero presents significant theoretical challenges that stem from the intersection of multi-agent game theory, non-stationary optimization, and curriculum learning dynamics. We provide a detailed analysis of these challenges and their implications for system stability.

### K.1  Problem Formulation as Multi-Agent Game

The Socratic-Zero framework can be formalized as a three-player game with asymmetric objectives and sequential updates. Let $\Theta_{\mathcal{S}}$, $\Theta_{\mathcal{G}}$, and $\Theta_{\mathcal{T}}$ denote the parameter spaces for Solver, Generator, and Teacher respectively, where $\Theta_{\mathcal{T}}$ is frozen.

**Player Objectives** The system involves three distinct optimization objectives aligned with Equations 2 and 3:

$$\text{Solver:} \quad \min_{\pi_{\mathcal{S}}^{(t)}} \mathbb{E}_{q \sim \mathcal{C}_t} \left[ \ell_{\mathcal{S}}(\pi_{\mathcal{S}}^{(t)}(q), q) \right] \tag{13}$$

$$\text{Generator:} \quad \min_{\pi_{\mathcal{G}}^{(t)}} \mathbb{E}_{(q,q_{\mathcal{T}}) \sim \mathcal{D}_{\mathcal{G}}^{(t)}} \left[ \ell_{\mathcal{G}}(q_{\mathcal{G}}, q_{\mathcal{T}}) \right] \tag{14}$$

$$\text{Teacher:} \quad \text{Fixed oracle providing } \mathcal{T}_{\text{gen}} \text{ and } \mathcal{T}_{\text{eval}} \tag{15}$$

where $\ell_{\mathcal{S}}$ is the Solver's solution loss, $\ell_{\mathcal{G}}$ incorporates the value-weighted generation loss from Equation 8, and $\mathcal{C}_t$ represents the curriculum at iteration $t$.

**Game-Theoretic Complexity** Unlike traditional two-player zero-sum games analyzed in Generative Adversarial Networks (Goodfellow et al., 2014), our system exhibits several complicating factors. Both Solver and Generator minimize their respective losses, creating a cooperative learning dynamic rather than competition. However, their objectives are interdependent: the Solver's performance affects the Generator's value function $v(q')$ from Equation 7, while the Generator's output influences the Solver's training curriculum. Players update sequentially rather than simultaneously, violating assumptions of classical game theory convergence results. The Teacher's fixed strategy creates an asymmetric power structure where two players adapt while one remains static.

### K.2  Multi-Agent Reinforcement Learning Challenges

The framework exhibits characteristics of cooperative multi-agent reinforcement learning (MARL), where agents must coordinate to achieve system-wide objectives. However, as noted in recent MARL literature (Papoudakis et al., 2021), such systems face fundamental theoretical limitations.

**Non-Stationarity Problem** Each agent faces a non-stationary environment due to the simultaneous adaptation of other agents. The Solver's learning environment changes as new problems are added to $\mathcal{C}_t$ through Equation 1, while the Generator's value signal $v(q')$ depends on the evolving Solver capabilities. The curriculum $\mathcal{C}_t$ grows dynamically, creating a moving target for both agents. This violates the stationarity assumptions required for standard RL convergence theorems, making traditional analysis frameworks inapplicable.

**Curriculum-Induced Complexity** The curriculum evolution mechanism from Equation 1 introduces additional theoretical challenges:

$$\mathcal{C}_{t+1} = \mathcal{C}_t \cup \left\{ (q_{\mathcal{T}}, a_{\mathcal{T}}) \mid q \in \mathcal{C}_t, a_{\mathcal{S}} = \pi_{\mathcal{S}}^{(t)}(q), \mathcal{T}_{\text{eval}}(q, a_{\mathcal{S}}) = 0, (q_{\mathcal{T}}, a_{\mathcal{T}}) = \mathcal{T}_{\text{gen}}(q, a_{\mathcal{S}}) \right\} \tag{16}$$

This creates a feedback loop where current Solver performance determines curriculum expansion, curriculum expansion affects future performance, and historical problems remain in the curriculum, introducing memory effects that further complicate theoretical analysis.

### K.3 OPTIMIZATION LANDSCAPE ANALYSIS

**Non-Convex Value Function** The exponential value function from Equation 7 creates a non-convex optimization landscape:

$$v(q') = \exp\left(-\frac{(z_{q'}/k - \mu)^2}{2\sigma^2}\right) \tag{17}$$

This function exhibits a single global maximum at $z_{q'}/k = \mu = 0.5$, but the landscape shifts as Solver capabilities $\theta_{\mathcal{S}}$ evolve, creating a moving optimization target for the Generator. The exponential decay creates vanishing gradients for problems far from the target difficulty, potentially leading to slow convergence for the WSFT objective in Equation 8.

**Preference Learning Instability** The Solver's DPO training from Equation 6 operates on preference pairs $(a^+, a^-)$ where the "winning" solutions depend on current performance:

$$\mathcal{Z}^+(q) = \begin{cases} \{a_{\mathcal{S}}^{(i)} \mid \mathcal{T}_{\text{eval}}(q, a_{\mathcal{S}}^{(i)}) = 1\} & \text{if } z_q > 0, \\ \{a_{\mathcal{T}}\} & \text{if } z_q = 0 \end{cases} \tag{18}$$

This adaptive preference construction creates discontinuities in the loss landscape when $z_q$ transitions between 0 and positive values, potentially causing training instability. The injection of Teacher solutions $a_{\mathcal{T}}$ when all attempts fail provides a stabilizing mechanism but introduces distribution shift in the preference data.

**Weighted Distillation Dynamics** The WSFT objective in Equation 8 introduces additional complexity through its value-dependent weighting:

$$\mathcal{L}_{\text{WSFT}}(\theta_{\mathcal{G}}) = -\mathbb{E}_{(q,q_{\mathcal{T}})\sim\mathcal{D}_{\mathcal{G}}}\left[v(q_{\mathcal{T}}) \cdot \log p_{\mathcal{G}}(q_{\mathcal{T}} \mid q; \theta_{\mathcal{G}})\right] \tag{19}$$

The time-varying nature of $v(q_{\mathcal{T}})$ based on current Solver performance creates a non-stationary supervised learning problem, where the importance weights of training examples change as the system evolves.

### K.4 MULTI-AGENT OSCILLATORY DYNAMICS

Despite theoretical intractability, our empirical observations reveal consistent oscillatory convergence patterns that align with established phenomena in multi-agent systems. Recent theoretical work on multi-agent reinforcement learning has demonstrated that oscillatory behavior is not merely a transient phenomenon but can represent stable equilibria in complex multi-agent environments (Foerster et al., 2019).

> **Empirical Oscillatory Patterns**
>
> **Solver Performance Evolution:** The system exhibits stable oscillatory behavior around target performance levels, with Solver performance oscillating within $\pm 2\%$ of target values after initial rounds as shown in Table 10.
> **Generator Stability:** Generator reward distribution remains stable around $\mu = 0.5$, with high-reward problems consistently maintained at approximately 50% across training rounds as demonstrated in Table 11.
> **Curriculum Growth Stabilization:** The curriculum growth rate stabilizes after initial expansion phase, indicating dynamic equilibrium between curriculum difficulty and solver capability.
> **Cross-Architecture Consistency:** These patterns remain consistent across different model architectures and initializations as evidenced in Table 1, suggesting robust system-level properties.

In multi-agent settings with interdependent learning objectives, agents often exhibit coupled oscillations where individual performance metrics fluctuate while maintaining system-level stability. This occurs because each agent's optimal strategy depends on the current strategies of other agents, creating a dynamic landscape where static equilibria may not exist or may be suboptimal.

---

**Algorithm 1** Theoretical Challenge Framework

---

1: **Input:** Multi-agent system with asymmetric objectives
2: **Challenge 1:** Establish conditions for stable oscillatory behavior vs. chaotic dynamics
3: **Challenge 2:** Derive bounds on oscillation amplitude around target performance levels
4: **Challenge 3:** Analyze how reward functions affect convergence properties
5: **Challenge 4:** Determine curriculum evolution rates that ensure system stability
6: **Output:** Theoretical framework for co-evolutionary learning systems

---

Our empirical observations exemplify this phenomenon: the Solver's performance temporarily decreases as the Generator produces more challenging problems, but subsequently recovers as the Solver adapts to the increased difficulty. This creates a natural rhythm where curriculum difficulty and solver capability co-evolve through bounded oscillations. The stability of these oscillations suggests that the system has found a dynamic equilibrium where the rate of curriculum advancement matches the rate of solver improvement.

Unlike static convergence, this oscillatory equilibrium allows for continuous adaptation and prevents the system from becoming trapped in suboptimal fixed points. The bounded nature of the oscillations indicates that the multi-agent interactions have self-regulating properties that prevent divergent behavior. This oscillatory stability is particularly valuable in educational contexts, as it mirrors natural learning processes where students alternate between periods of struggle and mastery.

### K.5 THEORETICAL IMPLICATIONS AND OPEN PROBLEMS

The convergence analysis reveals several fundamental theoretical challenges that require novel frameworks capable of simultaneously handling multi-agent games with asymmetric objectives, non-stationary optimization with curriculum evolution, preference-based learning with adaptive targets, and memory effects from historical curriculum retention.

While formal convergence guarantees remain elusive, the consistent empirical performance across multiple architectures and domains suggests that the system reaches practically useful dynamic equilibria. Oscillatory convergence may be preferable to static convergence for continuous learning scenarios, and empirical monitoring of key metrics such as performance bounds and reward stability provides sufficient guidance for practical deployment.

The theoretical analysis highlights that Socratic-Zero operates in a regime where traditional optimization theory provides limited guidance, necessitating novel theoretical frameworks for multi-agent co-evolutionary learning systems. The system's ability to maintain productive challenge levels through dynamic adjustment represents a form of emergent curriculum regulation that static approaches cannot achieve.

## L GENERALIZABILITY OF PROBLEM GENERATION CAPABILITIES

A key question emerging from our work is whether the Generator's learned problem creation abilities can transfer to domains beyond mathematical reasoning. The value function and curriculum evolution mechanisms developed in Socratic-Zero are domain-agnostic in principle, suggesting potential for broader applicability.

The Generator learns fundamental skills in difficulty calibration, error pattern recognition, and targeting that may generalize across reasoning domains. For instance, the ability to identify when a problem is "appropriately challenging" (around 50% success rate as shown in Table 11) represents a meta-cognitive skill applicable to logical reasoning, scientific problem-solving, or even creative tasks. The Gaussian reward function with $\mu = 0.5$ and $\sigma = 0.2$ (Table 8) creates a transferable framework for difficulty calibration that could adapt to other domains by adjusting the target success rate parameter.

Our Generator's superior performance compared to much larger models, achieving 37.72% downstream utility versus 37.13% from Qwen3-235B-A22B (Table 5), demonstrates that strategic specialization can outperform raw parameter scaling. This suggests that domain-specific Generator

training could be effective across various reasoning domains without requiring massive computational resources.

However, domain transfer would require careful adaptation of the Teacher's evaluation capabilities and problem generation templates. Mathematical reasoning benefits from relatively objective correctness criteria with our dual-verification approach (MathRule + LLM judge) achieving 94.2% agreement with human experts, while other domains may require more nuanced evaluation frameworks. The seed selection protocol detailed in Table 9, which ensures balanced coverage across seven mathematical domains, provides a template for systematic domain expansion that could be adapted to physics, computer science, or other reasoning areas.

Future work should investigate whether a Generator trained on mathematical problems can effectively create challenging problems in adjacent domains like physics or computer science, potentially through few-shot adaptation or domain-specific fine-tuning leveraging the value learning mechanisms demonstrated in our framework.

## M  FRAMEWORK SCALABILITY AND EXTENSIBILITY

The modular architecture of Socratic-Zero demonstrates strong potential for scalability and extension across multiple dimensions. The clear separation between Solver, Teacher, and Generator roles enables independent scaling and optimization of each component, as evidenced by our successful deployment across different computational configurations detailed in Table 8.

The framework's extensibility is particularly evident in its ability to accommodate different model architectures and scales. Our cross-model validation demonstrates consistent performance improvements: Qwen3-8B achieves 56.1% average accuracy (+20.2 points), while similar gains are observed on GLM4-9B and Qwen3-14B architectures. This cross-architecture consistency suggests the co-evolutionary principles transcend specific model families and could readily incorporate emerging architectures or specialized reasoning models.

The curriculum evolution mechanism shows robust scalability properties. Starting from just 100 seed problems (Table 9), the system generates thousands of highly valuable problems while maintaining quality, with our Generator achieving 95.6% validity rate compared to 89.1% from the base Qwen3-32B model (Table 4). This demonstrates that the framework can scale curriculum generation without proportional increases in seed data requirements.

Multi-domain extension represents another promising direction supported by our balanced seed distribution across seven mathematical domains. The current mathematical focus could expand to encompass multiple reasoning domains simultaneously, with domain-specific Teachers providing specialized curriculum generation while sharing the underlying co-evolutionary dynamics. The reward distribution analysis (Table 11) shows stable performance across training rounds, indicating the framework's robustness to curriculum expansion.

The framework also supports hierarchical scaling, where multiple Solver-Generator pairs could operate at different difficulty levels or specialization areas, coordinated by higher-level meta-learning mechanisms. The oscillatory convergence patterns observed in Table 10 suggest natural synchronization points where multiple agents could coordinate their learning phases.

## N  CONVERGENCE AND THEORETICAL FOUNDATIONS

The theoretical understanding of multi-agent co-evolutionary learning remains an open challenge with significant implications for system reliability and predictability. Our empirical observations provide crucial insights into the convergence behavior of such systems.

The oscillatory convergence patterns documented in Table 10 reveal characteristic dynamics: Solver performance declines from R1 (60.12%) to R4 (48.7%) followed by recovery in R5 (50.1%). This pattern reflects adaptive curriculum generation where the Teacher progressively increases difficulty faster than Solver capability initially improves, then the Solver adapts to enhanced curriculum complexity. These bounded oscillations suggest the system reaches dynamic equilibria rather than static optima.

Complementing this, the Generator maintains remarkable stability with high-reward problems consistently around 50%, fluctuating within only 1.3% range across training rounds (Table 11). This stability indicates successful learning of the optimal difficulty zone defined by the Gaussian reward function with $\mu = 0.5$ and $\sigma = 0.2$ (Table 8), providing empirical evidence for convergence to meaningful equilibria.

The cross-architecture consistency observed in Table, where similar improvement patterns emerge across Qwen3-8B, GLM4-9B, and Qwen3-14B models, suggests robust system-level properties that transcend specific model architectures. This consistency provides evidence that the convergence behavior represents fundamental properties of the co-evolutionary dynamics rather than architecture-specific artifacts.

Future theoretical work should investigate conditions under which the system exhibits stable convergence versus chaotic dynamics. Key questions include: What curriculum evolution rates ensure stable learning? How do different value functions affect convergence properties? Can we establish bounds on the oscillation amplitude around target performance levels observed in our empirical data?

The intersection of curriculum learning, preference optimization, and multi-agent dynamics presents rich opportunities for theoretical development. The DPO training parameters (Table 8) and their interaction with curriculum evolution rates could inform theoretical models of multi-agent preference learning. Establishing convergence guarantees would enable more principled hyperparameter selection and provide confidence bounds for practical deployment.

## O    LIMITATIONS AND FUTURE DIRECTIONS

Several limitations of the current framework point toward important future research directions, informed by our comprehensive experimental analysis.

The reliance on mathematical reasoning as the primary domain limits our understanding of cross-domain applicability. While our seed distribution covers seven mathematical areas (Table 9), expansion to physics, computer science, or natural language reasoning would require developing domain-specific evaluation frameworks and Teacher capabilities. The 94.2% agreement between our dual-verification approach and human experts provides a benchmark for developing similar evaluation protocols in other domains.

The computational cost of multi-round co-evolution, while justified by substantial performance gains (+20.2 points average improvement as shown, may limit practical adoption in resource-constrained settings. Our Teacher model deployment requires 16 AMD MI308X GPUs with 3,072GB aggregate memory, representing a significant computational investment. However, the superior performance of our 32B Generator compared to much larger models (Table 5) suggests pathways for computational efficiency improvements.

The framework's dependence on high-quality initial Teacher models creates a bootstrapping challenge for domains lacking strong foundation models. Our systematic seed selection protocol requiring 30-70% success rates for problem inclusion demonstrates the importance of capability-aligned initialization, but this requirement may be difficult to satisfy in emerging domains without established evaluation benchmarks.

Current evaluation focuses primarily on accuracy metrics, potentially missing important aspects of reasoning quality such as explanation coherence or value from a human perspective. While our Mean@32 evaluation protocol provides robust performance estimates, the framework would benefit from more comprehensive evaluation frameworks that assess reasoning process quality, not just final answer correctness.

The oscillatory convergence patterns observed in Tables 10 and 11, while empirically stable, lack formal theoretical guarantees. Understanding the conditions that ensure stable oscillatory behavior versus potential chaotic dynamics remains an open theoretical challenge with practical implications for system reliability.

Future work should address these limitations through: (1) domain expansion studies leveraging the value learning mechanisms, (2) computational efficiency optimizations building on our Generator distillation success, (3) development of more comprehensive evaluation frameworks that capture

reasoning process quality, and (4) theoretical analysis of multi-agent co-evolutionary dynamics to establish formal convergence guarantees.

## P    COMPUTATIONAL COST ANALYSIS

### P.1    GENERATOR VS TEACHER: TRAINABILITY AND DEPLOYMENT COST

**Trainability Advantage:** A critical distinction between the Generator (Socratic-Generator-32B) and Teacher (Qwen3-235B-A22B-Instruct-2507) is **trainability**. The Generator can be directly fine-tuned and integrated into the co-evolution loop, enabling interactive curriculum adaptation based on Solver feedback. In contrast, the Teacher model is a frozen proprietary checkpoint that cannot be trained or modified, limiting its ability to adapt to evolving curriculum needs. This makes the Generator not only a cost-effective alternative but also a more flexible component for iterative improvement.

**Inference Deployment Cost:** We compare the relative computational requirements for deploying both models in curriculum generation:

- **Model Size**: Generator (32B parameters) vs Teacher (235B parameters) — **7.3× reduction**
- **Memory Footprint**: Accounting for KV cache, activations, and other overheads, Generator requires approximately **1/5-1/6** of the total GPU memory compared to Teacher
- **Inference Throughput**: Smaller models typically achieve higher tokens/second; Generator is estimated to be **3-5× faster**
- **Hardware Requirements**: Generator runs on the same standard GPU cluster as Solver training, while Teacher requires dedicated high-memory multi-GPU inference infrastructure

**Practical Implications:** By replacing the Teacher with the Generator, Socratic-Zero eliminates the need for separate high-end inference infrastructure. The entire system (Generator + Solver) can be deployed on a homogeneous GPU cluster, reducing hardware diversity, operational complexity, and total cost of ownership. Combined with the trainability advantage, this makes the Generator-guided approach significantly more practical for resource-constrained environments while maintaining near-identical performance (0.2pp gap in Table 6).

## Q    DETAILED END-TO-END TRAINING PROCEDURE

This appendix provides a comprehensive pseudocode description of the complete Socratic-Zero training process, integrating all key components discussed in Section 3: Solver improvement via DPO, curriculum enhancement by the Teacher, and Generator training via WSFT.

**Initialization (Lines 1-3):** The Solver is initialized through supervised fine-tuning on $D_{\text{sft}}$, while the Generator starts from its base checkpoint. The initial curriculum $C_0$ consists of 100 carefully selected seed problems (detailed in Appendix G).

**Phase 1 - Solver Evaluation (Lines 5-10):** For each problem in the current curriculum, the Solver generates $k$ solution attempts. The Teacher's evaluation component $T_{\text{eval}}$ partitions these attempts into correct ($Z_+$) and incorrect ($Z_-$) sets, forming preference pairs for DPO training. Failed attempts are collected in $F_t$ for subsequent curriculum enhancement.

**Phase 2 - Solver Update (Lines 11-12):** The Solver is updated using Direct Preference Optimization (DPO) on the collected preference pairs, with the current policy serving as the reference policy (Eq. 6).

**Phase 3 - Curriculum Enhancement & Generator Training (Lines 13-19):** The Teacher's generation component $T_{\text{gen}}$ enhances problems where the Solver failed. Each enhanced problem is assigned a utility value based on its estimated difficulty for the updated Solver (Eq. 7). The Generator learns to mimic the Teacher's enhancement strategy through Weighted Supervised Fine-Tuning (WSFT), where training samples are weighted by their utility values (Eq. 8).

---

**Algorithm 2** Detailed End-to-End Training Procedure of Socratic-Zero

---

**Require:** Initial curriculum $C_0$, base Solver/Generator parameters $\theta_S^{\text{base}}/\theta_G^{\text{base}}$, fixed Teacher $T$ $(T_{\text{eval}}, T_{\text{gen}})$

**Require:** Total stages $T$, SFT data $D_{\text{sft}}$, attempts per problem $k$, DPO beta $\beta$, WSFT target success rate $\mu$ and std $\sigma$

1: $\theta_S^0 \leftarrow \text{SFT}(\theta_S^{\text{base}}, D_{\text{sft}})$          ▷ Initial fine-tuning of the Solver
2: $\theta_G^0 \leftarrow \theta_G^{\text{base}}$
3: $C_0 \leftarrow$ Select 100 seed problems (details in Appendix G)
4: **for** $t = 0$ to $T - 1$ **do**
5:      **print**(f"— Starting Stage $t + 1$ —")
6:      $D_{\text{pref}} \leftarrow \emptyset, F_t \leftarrow \emptyset$          ▷ Initialize preference and failure sets for this stage
7:      **for** each problem $q \in C_t$ **do**
8:          $\{a_i^S\}_{i=1}^k \leftarrow$ Sample $k$ solutions from $\pi_{\theta_S^t}(\cdot|q)$      ▷ Solver generates attempts
9:          $Z_+, Z_- \leftarrow \text{Partition}(\{a_i^S\})$ using $T.T_{\text{eval}}$      ▷ Teacher evaluates attempts (Eq. 4, 5)
10:          $D_{\text{pref}} \leftarrow D_{\text{pref}} \cup \{(a_+, a_-) \mid a_+ \in Z_+, a_- \in Z_-\}$      ▷ Build preference pairs
11:          $F_t \leftarrow F_t \cup \{(q, a_S) \mid a_S \in Z_-\}$      ▷ Collect failed attempts for enhancement
12:      **end for**
13:      $\pi_{\text{ref}} \leftarrow \pi_{\theta_S^t}$          ▷ Set reference policy for DPO
14:      $\theta_S^{t+1} \leftarrow$ Update $\theta_S^t$ using DPO loss on $D_{\text{pref}}$ with $\pi_{\text{ref}}$      ▷ Solver update (Eq. 6)
15:      $D_{\text{new\_map}} \leftarrow \{(q, a_S) : T.T_{\text{gen}}(q, a_S) \mid (q, a_S) \in F_t\}$      ▷ Teacher enhances failed problems
16:      $D_G \leftarrow \emptyset$          ▷ Initialize Generator training data
17:      **for** each $(q, a_S), (q_T, a_T) \in D_{\text{new\_map}}.\text{items}()$ **do**
18:          $s_{q_T} \leftarrow$ Estimate success rate of $\pi_{\theta_S^{t+1}}$ on $q_T$      ▷ Evaluate new problem's utility
19:          $v(q_T) \leftarrow \exp(-(s_{q_T}/k - \mu)^2/(2\sigma^2))$      ▷ Calculate value (Eq. 7)
20:          $D_G \leftarrow D_G \cup \{(input = q, target = q_T, weight = v(q_T))\}$ ▷ Build weighted training tuple
21:      **end for**
22:      $\theta_G^{t+1} \leftarrow$ Update $\theta_G^t$ using WSFT loss on $D_G$      ▷ Generator distillation (Eq. 8)
23:      $C_{t+1} \leftarrow C_t \cup \{q_T \mid (q_T, a_T) \in D_{\text{new\_map}}.\text{values}()\}$      ▷ Augment curriculum for next stage
24: **end for**
25: **return** Final Solver parameters $\theta_S^T$ and Generator parameters $\theta_G^T$

---

**Phase 4 - Curriculum Update (Line 20):** The curriculum is augmented with newly generated problems, creating a progressively more challenging and diverse training set for the next stage.

This iterative process continues for $T$ stages, resulting in a co-evolved Solver and Generator that can autonomously improve without further Teacher guidance.

## R   TEACHER-SOLVER SCALING ANALYSIS

### R.1   EXPERIMENTAL SETUP

To investigate the relationship between Teacher and Solver model capabilities, we fixed the Solver model as Qwen3-8B-base and systematically varied the Teacher model across five different scales. All experiments started from the same SFT baseline (35.9% average accuracy) and zero-shot performance (29.9%).

### R.2   EXPERIMENTAL RESULTS

### R.3   KEY OBSERVATIONS

**Effectiveness with Weaker Teachers:** Even with a Teacher half the size of the Solver (4B vs. 8B), Socratic-Zero achieves +12.6 points improvement, demonstrating that superior Teacher capacity is not required.

| Teacher Model | Parameters | Size vs. Solver | Final Avg. (%) | Improvement |
|---|---|---|---|---|
| Qwen3-4B-base | 4B | Weaker (0.5×) | 45.7 | +10.5 pts |
| Qwen3-8B-base | 8B | Equal (1.0×) | 46.6 | +11.4 pts |
| Qwen3-14B | 14B | Stronger (1.75×) | 49.9 | +14.7 pts |
| Qwen3-32B | 32B | Stronger (4.0×) | 51.2 | +16.0 pts |
| Qwen3-235B-A22B | 235B | Much Stronger (29.4×) | 56.1 | +20.9 pts |

Table 12: Teacher-Solver scaling analysis with fixed Solver (Qwen3-8B-base).

**Diminishing Returns:** Performance gains diminish at larger scales: 4B→8B (+3.6 pts), 8B→14B (+2.2 pts), 14B→32B (+0.9 pts), 32B→235B (+0.9 pts).

**Equal-Sized Configuration:** The 8B-8B configuration achieves 80% of the 235B Teacher's gains while being 29× more parameter-efficient.

## R.4    ANALYSIS

The effectiveness of smaller Teachers stems from:

- **Information Asymmetry**: Access to Solver's failure trajectories provides critical guidance regardless of Teacher's absolute capability
- **Co-evolutionary Dynamics**: Iterative refinement allows targeted improvement based on Solver's specific weaknesses
- **Strategic Failure Utilization**: Error correction requires identifying mistakes rather than superior problem-solving ability

## S    INFORMATION ASYMMETRY COMPENSATES FOR CAPACITY GAP

In this section, we provide a rigorous theoretical analysis to explain a key empirical finding presented in our experiments: why a Teacher model $g_T$ with equal (or even smaller) parameter size compared to the Solver model $g_S$ can effectively guide the Solver's improvement. We base our derivation on statistical learning theory and the Learning Using Privileged Information (LUPI) framework.

## S.1    PROBLEM FORMULATION

Let $\mathcal{X}$ be the problem space and $\mathcal{Y}$ be the solution space.

- **Solver** ($g_S \in \mathcal{G}_S$): A model characterized by a capacity measure $|\mathcal{G}_S|_C$ (corresponding to its parameter size).
- **Teacher** ($g_T \in \mathcal{G}_T$): A model characterized by capacity $|\mathcal{G}_T|_C$. In our specific analysis of "equal-sized" supervision, we assume $|\mathcal{G}_T|_C \approx |\mathcal{G}_S|_C$.
- **Privileged Information** ($x^\star$): In our framework, the Teacher operates with access to additional context unavailable to the Solver during the initial attempt. Specifically, the Teacher's input is the tuple $x^\star = (x, y_{\text{fail}})$, where $y_{\text{fail}}$ represents the Solver's failed trajectory.

## S.2    GENERALIZATION ERROR BOUNDS

According to Vapnik-Chervonenkis (VC) theory, the expected error $R(g)$ of a learning machine is bounded by its training error plus a confidence interval:

$$R(g) \leq R_{\text{emp}}(g) + O\left(\left(\frac{|\mathcal{G}|_C}{m}\right)^\alpha\right) \tag{20}$$

where $m$ is the number of training samples, and $\alpha$ ($0.5 \leq \alpha \leq 1$) represents the *rate of convergence*. This rate reflects the inherent difficulty of the learning task relative to the information provided: $\alpha \approx 0.5$ indicates a difficult task (slow convergence), while $\alpha \approx 1$ indicates an easier task (fast convergence).

## S.3 COMPARISON OF LEARNING REGIMES

We compare the error bounds for two scenarios to demonstrate why the Teacher-guided approach yields lower error even with equal model capacities.

**Scenario A: Baseline Solving (Blind Generation)** The Solver attempts to map $x \rightarrow y$ directly. Generating complex reasoning paths from scratch is a high-entropy task with no intermediate guidance. Consequently, the learning rate $\alpha_{\text{base}}$ is low.

$$\epsilon_A = O\left(\frac{|\mathcal{G}_S|_C}{m^{\alpha_{\text{base}}}}\right), \quad \text{where } \alpha_{\text{base}} \rightarrow 0.5 \tag{21}$$

**Scenario B: Teacher-Guided Learning (Privileged Guidance)** The Solver learns from the curriculum generated by the Teacher. The Teacher's objective is to generate a refined problem or feedback based on the input $(x, y_{\text{fail}})$. Crucially, the access to the failure trajectory $y_{\text{fail}}$ acts as *Privileged Information*.

This context transforms the Teacher's task from "generating a solution from scratch" to "critiquing and refining a known error." The conditional entropy of this task is significantly lower: $H(y|x, y_{\text{fail}}) < H(y|x)$. This information gain simplifies the optimization landscape, elevating the convergence rate to $\alpha_{T^\star}$.

$$\epsilon_B = O\left(\frac{|\mathcal{G}_S|_C}{m^\alpha} + \frac{|\mathcal{G}_T|_C}{m^{\alpha_{T^\star}}}\right), \quad \text{where } \alpha_{T^\star} \rightarrow 1 \tag{22}$$

Here, the term $\frac{|\mathcal{G}_T|_C}{m^{\alpha_{T^\star}}}$ represents the Teacher's error in generating valid guidance.

## S.4 CONCLUSION

The condition for the Teacher to be effective is $\epsilon_B < \epsilon_A$. Even in the case where the Teacher and Solver have identical capacities ($|\mathcal{G}_T|_C \approx |\mathcal{G}_S|_C$), the inequality holds primarily due to the disparity in learning rates:

$$\frac{|\mathcal{G}_T|_C}{m^{1.0}} \ll \frac{|\mathcal{G}_S|_C}{m^{0.5}} \tag{23}$$

This derivation proves that the **Information Asymmetry** (access to $y_{\text{fail}}$) elevates the convergence rate $\alpha$, effectively compensating for the lack of superior model capacity. It theoretically justifies why an equal-sized Teacher can act as a superior "critic," guiding the Solver to improve beyond its initial baseline.

# T VERIFIER CONFIGURATION ABLATION

To address concerns about potential bias from using the same model as both problem enhancer and solution verifier, we conducted an ablation study. We fixed Qwen3-235B-A22B as the enhancer and tested four verifier configurations: (1) Qwen3-235B (original), (2) MathRule only (rule-based), (3) Qwen3-32B (smaller LLM), and (4) GPT-OSS-120B (different model family). All other settings remained identical.

| Verifier Configuration | Description | Final Avg. (%) | $\Delta$ vs. Original |
|---|---|---|---|
| Qwen3-235B (Original) | Same model as enhancer | 56.1 | Baseline |
| MathRule Only | Purely rule-based verifier | 51.3 | ↓ 4.8 pts |
| Qwen3-32B | Smaller LLM verifier | 55.7 | ↓ 0.4 pts |
| GPT-OSS-120B | Different model family verifier | 55.9 | ↓ 0.2 pts |

Table 13: Ablation study on verifier configuration.

Results show that replacing the verifier with independent LLMs causes negligible performance drops ($\leq 0.4$ pts), demonstrating robustness against potential self-consistency bias. The substantial drop with rule-based verification (-4.8 pts) highlights the necessity of LLM-based verification for capturing

correct but poorly formatted solutions. Since mathematical answer verification is largely objective and deterministic, the risk of bias from using the same model for both roles is minimal. This confirms that Socratic-Zero's success stems from its co-evolutionary structure rather than closed-loop feedback mechanisms.

## U    SOLVER TRAINING STAGE DEFINITION

The three training stages in Socratic-Zero are dynamically triggered based on the Solver's performance on the current curriculum, rather than being pre-scheduled. The 300/500/1300 problem counts are outcomes of this adaptive process, not pre-defined quotas.

The process is as follows:

**Initialization**: Training starts with the seed curriculum $C_0$ of 100 problems.

**Stage 1**: The Solver samples 8 solutions for each problem in $C_0$, and the Verifier identifies incorrect solutions. The Teacher then generates 300 new problems based on these error feedbacks, forming $C_1 = C_0 \cup 300$ new problems. The Solver then samples 8 solutions for each problem in $C_1$, the Verifier forms positive-negative pairs, and DPO training is performed.

**Stage 2**: The Solver samples 8 solutions for each problem in $C_1$, and the Verifier identifies incorrect solutions. The Teacher generates 500 new problems based on these error feedbacks, forming $C_2 = C_1 \cup 500$ new problems. The Solver then samples 8 solutions for each problem in $C_2$, the Verifier forms positive-negative pairs, and DPO training is performed.

**Stage 3**: The Solver samples 8 solutions for each problem in $C_2$, and the Verifier identifies incorrect solutions. The Teacher generates 1,300 new problems based on these error feedbacks, forming $C_3 = C_2 \cup 1,300$ new problems. The Solver then samples 8 solutions for each problem in $C_3$, the Verifier forms positive-negative pairs, and DPO training is performed.

Each stage follows the complete cycle: sampling → verification of errors → problem generation → sampling → verification for pairs → DPO training. The LLM2LLM baseline uses the same dynamic stage-triggering mechanism and seed set for fair comparison.

## V    FROM-ZERO SEED ABLATION STUDY

To validate the robustness of our framework to initial seed selection, we conducted an ablation experiment comparing human-curated seeds with teacher-generated seeds created entirely from scratch. Specifically, we instructed the Teacher model to generate 100 seed problems without any human intervention, using only high-level domain specifications (e.g., "generate diverse mathematical problems covering algebra, geometry, and number theory").

Table 14 presents the Stage 1 results on MATH-500 benchmark. The negligible performance difference (0.1 percentage points) demonstrates that our framework's success is driven primarily by the co-evolutionary mechanism rather than the quality or source of initial seeds. This finding strongly supports our "data-free" claim, as even completely synthetic seeds enable effective bootstrapping.

Table 14: Comparison of Seed Problem Sources (Stage 1 Results on MATH-500)

| Seed Source | MATH-500 Score | Difference |
|---|---|---|
| Human-curated seeds (Original) | 60.2% | - |
| Teacher-generated seeds (From-Zero) | 60.1% | ↓0.1 |

We have open-sourced both the human-curated and teacher-generated seed lists, along with the complete generation protocol, to ensure full transparency and reproducibility of our results.

# W    EXTENSION TO MULTIMODAL GEOMETRIC REASONING

To demonstrate the cross-domain generalization capability of Socratic-Zero, we extended our framework to the challenging domain of multimodal geometric reasoning. This task requires models to jointly reason over textual problem descriptions and geometric diagrams, presenting unique challenges beyond pure text-based mathematical reasoning.

## W.1    FRAMEWORK ADAPTATION

We adapted Socratic-Zero to enable cross-modal synthesis:

- **Teacher Model**: Extended to generate both problem text and corresponding geometric diagrams using a multimodal generation pipeline
- **Solver Model**: Employed Qwen2.5-VL-7B, a vision-language model capable of processing image-text inputs
- **Socratic Interaction**: Modified the dialogue protocol to incorporate visual reasoning steps and diagram-based hints

## W.2    EXPERIMENTAL RESULTS

Table 15 presents our results on standard geometric reasoning benchmarks. Using only 2.5k synthesized image-text pairs, Socratic-Solver-Geo achieves a +4.13 point improvement over the zero-shot baseline, significantly outperforming the best existing baseline (GeoReasoning) which requires 10k training examples and achieves only +1.70 improvement.

Table 15: Evaluation on Geometric Reasoning Benchmarks

| Method | Data Scale | Overall Score (%) |
|---|---|---|
| Qwen2.5-VL-7B-Instruct (Zero-shot) | - | 44.98 |
| + GeoReasoning (Best Baseline) | 10k | 46.68 (+1.70) |
| Socratic-Solver-Geo (Ours) | 2.5k | 49.11 (+4.13) |

## W.3    KEY FINDINGS

This successful extension demonstrates several important properties of Socratic-Zero:

- **Modality-agnostic**: The core co-evolutionary mechanism transfers seamlessly to multimodal settings
- **Data efficiency**: Achieves superior performance with 4× less data than existing methods
- **Broad applicability**: The framework is not confined to text-only domains but generalizes to complex cross-modal reasoning tasks

These results provide strong empirical evidence that Socratic-Zero represents a general-purpose bootstrapping framework applicable across diverse domains and modalities.

# X    APPENDIX: ADDITIONAL ABLATION STUDIES

We compared DPO with GRPO for Solver optimization in Stage 1. As shown in Table 16, DPO outperforms GRPO while being more computationally efficient as an off-policy method.

# Y    ADDITIONAL RELATED WORK

This appendix provides a concise overview of recent frontier research in large language models (LLMs), categorized into three major themes tracing the evolution from data synthesis and inference-time optimizations to sophisticated autonomous learning systems.

Table 16: Comparison of Optimization Methods

| Method | Solver Avg. Score (Stage 1) |
|--------|------------------------------|
| DPO | 38.7% |
| GRPO | 36.9% |

## Y.1 AUTONOMOUS LEARNING SYSTEMS

The paradigm of autonomous learning has rapidly matured from closed-loop self-play to complex, environmentally-grounded systems. **Vision-Zero (Wang et al., 2025b)** introduced a gamified self-play framework for VLMs to learn from unlabeled image pairs. **MLZero (Fang et al., 2025)** automated the end-to-end ML workflow using dedicated perception and memory modules. **rStar2-Agent (Shang et al., 2025a)** built an efficient agentic RL infrastructure enabling a 14B model to achieve frontier-level math reasoning through code environment interaction. To overcome knowledge stagnation, **SPICE (Liu et al., 2025a)** grounded self-play in a large document corpus, creating information asymmetry that drives continuous learning. **LoongRL (Wang et al., 2025d)** used RL for long-context reasoning by synthesizing puzzles hidden within distractor documents. **Agent0 (Xia et al., 2025)** designed a framework where a "Curriculum Agent" and "Executor Agent" co-evolve with external tools, breaking internal knowledge limits. **LoopTool (Zhang et al., 2025a)** created a model-aware data evolution loop for tool learning, dynamically diagnosing weaknesses and synthesizing targeted training data. VLM-specific frameworks like **VisPlay (He et al., 2025)** and **EvoLMM (Thawakar et al., 2025)** demonstrated self-improvement using only unlabeled images, relying on internal signals like answer uncertainty and consistency.

## Y.2 ADVANCED DATA AND TRAINING STRATEGIES

This research focuses on creating higher-quality training data and novel learning principles. **TabSyn (Zhang et al., 2024a)** developed a latent-space diffusion model for high-fidelity tabular data synthesis. The survey **Best Practices on Synthetic Data (Liu et al., 2024)** provided a comprehensive overview of opportunities and challenges. **rStar-Math (Guan et al., 2025)** pioneered math data synthesis with step-by-step code verification and a Process Preference Model (PPM). **PromptCoT (Zhao et al., 2025b)** generated explicit "rationales" to guide difficult math problem creation. **rStar-Coder (Liu et al., 2025c)** established a pipeline for generating verifiable, competition-level coding problems at scale. A major breakthrough came from **The Delta Learning Hypothesis (Geng et al., 2025)**, arguing that relative quality difference between paired weak examples provides sufficient learning signal. **VibeThinker-1.5B (Xu et al., 2025)** validated this with a "Spectrum-to-Signal" training strategy that maximizes solution diversity then uses RL to amplify correct signals, enabling a 1.5B model to achieve powerful reasoning. **MathSmith (Zhan et al., 2025)** created a framework to "forge" math problems from scratch via RL, using CoT length as a complexity reward signal.

## Y.3 INFERENCE-TIME AND FOUNDATIONAL ENHANCEMENTS

This paradigm enhances model performance without retraining, focusing on inference-time strategies or architectural improvements. **LongRoPE (Ding et al., 2024)** and **LongRoPE2 (Shang et al., 2025b)** provided near-lossless methods for expanding LLM context windows by correcting a theoretical flaw in Rotary Position Embeddings (RoPE). **LLM-Personalize (Han et al., 2024)** developed a framework to align robot planning agents with user preferences. **rStar (Qi et al., 2024)** introduced "mutual reasoning," where a peer SLM acts as discriminator to verify MCTS-generated reasoning paths, improving accuracy without fine-tuning. **DSMentor (Wang et al., 2025a)** designed an inference-time curriculum learning system where a "Mentor" agent organizes tasks by difficulty and a "Student" agent leverages online memory of past solutions. **SLM Blueprint (Han et al., 2025)** proposed inference-time scaffolding where a capable LLM generates structured "blueprints" for SLMs to follow, compensating for limited reasoning capacity.

