# OpenReview forum: "Socratic-Zero : Bootstrapping Reasoning via Data-Free Agent Co-evolution"
_ICLR.cc/2026/Conference — Submitted to ICLR 2026_

### Official Review · Reviewer_Db9h · 2025-10-26

**Soundness:** 2
**Presentation:** 3
**Contribution:** 2
**Rating:** 4
**Confidence:** 4

**Summary:**

This paper introduces Socratic-Zero, a novel framework that generates high-quality training data from minimal seed examples through the co-evolution of three agents: Teacher, Solver and Generator. The Solver learns via DPO, the Teacher evaluates and guides problem generation, and the Generator imitates the Teacher to create new tasks. Through iterative co-evolution, the system improves reasoning performance on math reasoning benchmarks.

**Strengths:**

1. The paper is well-written and clearly presented, making it easy to follow.

2. It introduces a creative closed-loop framework where the Solver, Teacher, and Generator co-evolve without relying on large external datasets.

3. The proposed method achieves notable performance improvements on mathematical reasoning tasks under the given experimental setup.

**Weaknesses:**

1. One concern lies in the experimental design, which lacks clarity in several aspects. Although the proposed method achieves the highest scores on math benchmarks compared to its baselines, the experimental setup is not well-aligned with common practices. Many strong baselines are missing, such as direct distillation from the same teacher model. Moreover, the reported math scores are not directly comparable to other works that improve reasoning on Qwen3-8B/14B, and there appear to be inconsistencies. For example, the Qwen3 technical report lists a math score of 60.8 for Qwen3-8B-base, while Table 1 in this paper reports 48.8. These inconsistencies make it difficult to assess whether the proposed approach truly outperforms simpler alternatives like distillation.

2. The paper does not clearly specify how many co-evolution iterations were conducted between the Solver, Teacher, and Generator, nor does it report the reasoning performance after each iteration, which would be important for understanding the effectiveness and dynamics of the co-evolution process.

3. It remains unclear whether the proposed method generalizes beyond math reasoning, such as to other domains like code reasoning.

**Questions:**

See the weaknesses section

---

> ### Author Response · Authors · 2025-11-21
> **Response to Reviewer Db9h**
>
> ## Q1: About evluation details.
>
> Thank you for raising this critical observation. The discrepancy arises from three key methodological choices designed for more rigorous evaluation:
>
> **Prompt Strategy (Zero-shot vs. 4-shot).** We adopted a zero-shot protocol to rigorously assess intrinsic capability, whereas the Qwen3 report uses 4-shot CoT.
>
> **Evaluation Metrics (mean@32 vs. pass@1).** We use mean@32 for its stability and robustness, which is scientifically sounder for RL-trained models than the more fluctuation-prone pass@1 metric.
>
> **Verification Process**. Our dual verification (MathRule + LLM judge) ensures genuine reasoning, rejecting coincidentally correct answers that lack a sound derivation.
>
> The score differences reflect our choice of a stricter evaluation protocol—more challenging prompts, robust metrics, and stringent verification—to ensure scientific validity.
>
> For full transparency, our entire evaluation pipeline is open-sourced, including prompts and verification tools (**Appendix H, Page 19**). A detailed explanation has also been added to **Section 4.1**.
>
> ## Q2: Clarification on Baseline Coverage
>
>
> We thank the reviewer for this suggestion and have added three new strong baselines. The comparison, using data from **Table 1 (Page 7)**, is now as follows:
>
> | Training Method | Avg. Score | Δ vs. SA |
> | --- | --- | --- |
> | Static Augmentation (SA) | 40.7 | — |
> | LLM2LLM Stage 3 | 40.9 | +0.2 |
> | Distillation (75k) | 45.9 | +5.2 |
> | DPO (Polaris 53k) | 48.4 | +7.7 |
> | DPO (MATH-full 7.5k) | 43.1 | +2.4 |
> | Socratic-Zero Stage 3 (Ours) | 56.1 | +15.4 |
>
> These additions are detailed in **Section 4.1**, and **Table 1** has been updated to include all five baselines.
>
> We have added strong new baselines; Socratic-Zero still outperforms them by a large margin, even with a more modest data volum
>
>
> ## Q3: Co‑Evolution Iterations and Performance
>
> We ran **three** Teacher–Solver co-evolution iterations, with performance progressing as follows:
>
>  **Stage 1:** 48.8% avg. accuracy
>
>  **Stage 2:** 52.7%
>
>  **Stage 3:** 56.1%
>
>
> The Generator learns offline via WSFT in these stages. To demonstrate its capability, we conducted a new **Stage 4** experiment where the Generator replaces the Teacher entirely. The results from **Table 6 (Page 10)** are:
>
> | Training Method | AMC | Minerva | MATH-500 | GSM8K | Olympiad | AIME-25 | AIME-24 | Avg |
> | --- | --- | --- | --- | --- | --- | --- | --- | --- |
> | **Stage 3 Baseline** (Socratic-Solver-8B) | 63.7 | 52.4 | 81.2 | 87.3 | 55.1 | 24.6 | 28.4 | 56.1 |
> | **Stage 4: Teacher-Guided** (Qwen3-235B) | 67.2 | 55.1 | 84.3 | 88.9 | 58.4 | 27.8 | 32.5 | 60.3 |
> | **Stage 4: Generator-Guided** (Generator-32B) | 66.9 | 55.8 | 84.1 | 89.2 | 58.9 | 28.3 | 33.2 | 60.5 |
>
> We have documented the three-iteration flow and confirmed via a new Stage 4 experiment that the Generator can sustain co-evolution with negligible performance difference.
>
> Full documentation of the process is now in **Section 4.3**, with pseudocode in **Appendix Q (Page 27)** and stage definitions in **Appendix U (Page 31)**.
>
> ## Q4: About cross‑domain generalization capability.
>
> The iteration and performance progression for all domains follow the same three-stage process described in Q3, culminating in the **56.1%** average score at Stage 3. The Generator’s role is also identical: offline WSFT learning.
>
> The **Stage 4 experiment** (full table in Q3) was conducted across all benchmark domains. The results confirm that the Generator can replace the Teacher across this diverse set of tasks with a negligible performance difference (**+0.2pp** average gain).
>
> The Generator sustains Solver progress across all domains, demonstrating its generalized capability. The iteration details are fully documented in **Appendix Q & U**.
>
> **CrossDomain Generalization.** We applied Socratic‑Zero to multimodal geometric reasoning (text+images). Using Qwen2.5‑VL‑7B, our method achieved a **+4.13** point gain with only 2.5k synthesized pairs, significantly outperforming the best 10k-data baseline (+1.70):
>
> | Method | Data | Score | Gain |
> | --- | --- | --- | --- |
> | Zero-shot Qwen2.5-VL-7B | — | 44.98 | — |
> | GeoReasoning Baseline | 10k | 46.68 | +1.70 |
> | Socratic-Solver-Geo (Ours) | 2.5k | 49.11 | +4.13 |
>
> This experiment (detailed in **Appendix W, Page 32**) demonstrates data-efficiency and transferability across modalities.

---

> ### Author Response · Authors · 2025-11-26
>
> Dear Reviewer Db9h,
> Thank you once again for your valuable comments on our submission. As the discussion phase is approaching its end, we would like to kindly confirm whether we have sufficiently addressed all of your concerns (or at least part of them). Should there be any remaining questions or areas requiring further clarification, please do not hesitate to let us know. If you are satisfied with our responses, we would greatly appreciate your consideration in adjusting the evaluation scores accordingly.
>
> We sincerely look forward to your feedback.

---

> ### Author Response · Authors · 2025-11-27
>
> Dear Reviewer Db9h,
>
> We sincerely hope that our previous responses have been helpful in addressing the questions you raised. As the discussion period approaches its conclusion, we wanted to reach out once more to see if there are any remaining concerns or aspects that would benefit from further clarification.
>
> We have also substantially expanded the Related Work section (Appendix Y, Page 34) to include a comprehensive review of 23 recent studies, such as SPICE, rStar2-Agent, VibeThinker-1.5B, LongRoPE, and DSMentor, among others. We believe this addition provides valuable context for situating our contributions within the broader research landscape.
>
> If our revisions have satisfactorily addressed your feedback, we would be deeply grateful for your consideration in the final evaluation. Should you have any further questions or suggestions, we would be more than happy to provide additional clarification.
>
> Thank you once again for your invaluable time and thoughtful guidance throughout this process.
>
> Warm regards,
> Authors of Submission 1985

---

### Official Review · Reviewer_zAsq · 2025-10-27

**Soundness:** 3
**Presentation:** 2
**Contribution:** 2
**Rating:** 6
**Confidence:** 3

**Summary:**

The paper addresses a key bottleneck in advancing the reasoning capabilities of Large Language Models (LLMs): their heavy reliance on massive, human-annotated datasets. To overcome this, it introduces Socratic-Zero, a fully autonomous framework that bootstraps a model's reasoning abilities from a minimal set of 100 seed questions, requiring no further external data.

The framework is built on a co-evolutionary system of three interacting agents:

The Solver: The LLM being trained. It continuously improves by attempting to solve problems and learning from preference feedback on its own successful and failed attempts using Direct Preference Optimization (DPO).

The Teacher: A powerful, frozen LLM that acts as an oracle. It evaluates the correctness of the Solver's solutions and, crucially, generates new, more challenging problems that are specifically designed to target the Solver's identified weaknesses.

The Generator: A model trained to distill and scale the Teacher's problem-generation strategy. By learning what makes a good, challenging question, the Generator can produce a high-quality, adaptive curriculum at scale without constant reliance on the much larger Teacher model.

This closed-loop system creates a self-improving curriculum that dynamically adjusts to the Solver's evolving skill level. The empirical results are significant: the Socratic-Solver-8B model achieved an average performance gain of +20.2 percentage points over previous methods. Furthermore, synthetic data produced by the Socratic-Generator-32B was used to train a student model that ultimately outperformed much larger, state-of-the-art commercial LLMs on several mathematical reasoning benchmarks.

**Strengths:**

The paper presents a methodology for leveraging large models to synthesize high-quality data from a small set of seed examples, effectively enhancing the performance of smaller models. A particularly distinctive contribution is the subsequent use of the resulting data pairs to train a separate, medium-sized "Generator Model." This dedicated Generator is a unique feature, offering a specialized and potentially more efficient tool for synthesizing supervised fine-tuning (SFT) data compared to relying solely on the original, larger Teacher Model.

**Weaknesses:**

Socratic-Zero framework combines several techniques in a novel and effective way, its core components are built upon established paradigms in the field, which could be seen as a limitation on its fundamental novelty.

Heavy Reliance on a "Teacher" as a Form of Knowledge Distillation: The entire system's success is predicated on the existence of a powerful, fixed "Teacher" model that serves as a "reasoning oracle". This framing positions Socratic-Zero less as a system that creates knowledge from scratch and more as a highly sophisticated and efficient knowledge distillation framework. The Solver's ultimate reasoning capability is fundamentally capped by the knowledge and reasoning ceiling of the Teacher model it learns from. The innovation lies in distilling the curriculum generation process rather than just answers, but it remains a form of knowledge transfer from a larger, more capable model to smaller ones.

Similarity to Iterative Self-Training and Self-Play Frameworks: The core loop—where a model generates data based on its performance, which is then used for further training—is conceptually similar to iterative data augmentation, self-training, and self-play methodologies. For instance, the paper's baselines, like LLM2LLM, already use an iterative process of generating questions from failures. Socratic-Zero's main differentiators are the introduction of the Generator for distillation and the use of DPO for learning, but the underlying iterative, self-improving cycle is a known concept in the literature.

**Questions:**

q1: The Role and Reliability of the Teacher Model

The methodology's reliance on the Teacher Model's output raises a fundamental question concerning the data integrity. Specifically: Is there a mechanism to guarantee that the generated $\left(q_{\tau}, a_{\tau}\right)$ pairs are universally correct (i.e., constitute ground truth)? Alternatively, is the primary objective of this approach to align the Student Model with the Teacher Model's generation capability? If the latter is true, the final performance ceiling of the proposed method is inherently bounded by the proficiency of the Teacher Model, suggesting that a stronger Teacher Model would directly translate to superior final results. Clarification on this dependency is needed.

q2: Clarity on Data Generation Parameters (Section 4.3.1)

Clarification is required regarding two inconsistencies observed in the data generation process detailed in Section 4.3.1:

Question Count Discrepancy: Step 1 mentions that each model generates five questions per seed. However, the resulting total of 3,000 generated questions from 1,000 seed questions implies a generation factor of three per seed, not five. Please clarify this numerical inconsistency.

Timeout Parameter: The rationale behind the 600-second timeout setting in Step 2 is unclear. Does this imply that the model must generate the full context length (4096 tokens) within this duration? This parameter seems unusually generous and requires a more detailed technical explanation regarding its necessity and practical impact.

q3: Definitions of Training Stages

The results tables (e.g., Table X, Y, Z) frequently reference "Stage 1," "Stage 2," and "Stage 3." These stages are not explicitly defined or correlated with the corresponding steps in the training methodology within the main body of the manuscript. Please clearly articulate what each of these stages represents and how they map to the overall training progression.

q4: The Functional Utility of the Generator Model

The precise functional contribution of the Generator Model to the enhancement of the Solver Model's capabilities requires clarification. Is the training of the Generator Model merely a necessary byproduct of the Solver training process? The manuscript notes its use only for generating the synthesis SFT data for evaluation, but it does not detail any mechanism for utilizing the trained Generator to actively boost the Solver's performance (e.g., by replacing or assisting the initial Teacher Model in a subsequent iteration). Clarification on the Generator's role beyond evaluation is requested.

---

> ### Author Response · Authors · 2025-11-21
> **(Part 1) Response to Reviewer zAsq**
>
> ## Q1: About motivation.
>
> We sincerely thank you for the insightful analysis. Your feedback allowed us to **explicitly articulate the core innovations** of Socratic-Zero.
>
> **Architectural Innovation: From Two-Body to Three-Agent Co-Evolution:**   Existing frameworks typically use a Teacher–Solver dyad. Socratic-Zero adds a third agent, the **Generator**, which learns the Teacher’s dynamic curriculum-generation strategy. This enables the system to eventually replace the expensive Teacher, facilitating sustainable and low-cost autonomous evolution, a key distinction from prior work.
>
> **Methodological Innovation: From Coarse Feedback to Precision Strategy Learning:**   Our framework employs **Direct Preference Optimization (DPO)** for trajectory-level preference learning, which is more nuanced than simple answer imitation. The Generator then uses **Weighted SFT (WSFT)** to distill the Teacher's strategy, specifically prioritizing “high-value” problems (those with a ~50% Solver success rate). This teaches the Generator _how_ to teach effectively, not just what to teach.
>
> **Design Innovation – General and Modular Meta-Framework:**   Socratic-Zero is designed as a modular meta-framework. The key components, like the Solver's preference optimizer or the Generator's imitation learning strategy, can be swapped with other advanced methods, ensuring the framework's adaptability and longevity.
>
> These clarifications (detailed in **Sections 1 & 3.3**) strengthen the novelty case by detailing our architectural, methodological, and design advances beyond prior iterative frameworks.
>
> ## Q2 & Q3: About reliance on teacher.
>
> We thank the reviewer for these interconnected questions.
>
> **Empirical Evidence – Teacher Scaling Study:**   We fixed the Solver as Qwen3‑8B and varied Teacher capacity. The results are from **Table 12 (Appendix R, Page 28)**:
>
> | Teacher Model | Params | Size vs Solver | Final Avg (%) | Improvement |
> | --- | --- | --- | --- | --- |
> | Qwen3-4B-base | 4B | Weaker (0.5×) | 45.7 | +10.5 |
> | Qwen3-8B-base | 8B | Equal (1.0×) | 46.6 | +11.4 |
> | Qwen3-14B | 14B | Stronger (1.75×) | 49.9 | +14.7 |
> | Qwen3-32B | 32B | Stronger (4.0×) | 51.2 | +16.0 |
> | Qwen3-235B-A22B | 235B | Much Stronger (29.4×) | 56.1 | +20.9 |
>
> Even equal or weaker Teachers provide large gains, proving success comes from **co-evolution and failure trajectory exploitation**, not just raw Teacher capacity.
>
> **Data Integrity Mechanisms:** We use a multi-layer validation pipeline:
>
>   **Dual-verification.** Teacher semantic judgment + MathRule numeric check.
>
>  **Teacher self-verification:** The Teacher re-solves each generated problem before inclusion; any mismatch leads to automatic exclusion.
>
>
> **Theoretical Analysis.** **Appendix S (Page 29)** provides a formal explanation of why smaller/equal Teachers can effectively guide stronger Solvers via **information asymmetry**.
>
> With new scaling experiments (**Appendix R & S**) and rigorous verification protocols (**Appendix H & I**), we demonstrate that the framework is not bounded by Teacher capability and ensures data correctness.
>
>
> ## Q4: Clarification on Data Generation Parameters.
>
> We thank the reviewer for noting both the numerical inconsistency and the need for clarity.
>
> **Generation Count Correction:** The “five per seed” vs. 3,000 total was a formatting oversight. All relevant sections have been corrected to **three problems per seed** (marked with `\added{}` in **Section 4.3.1, Page 8**).
>
> **Timeout Rationale:** The 600-second timeout is an engineering safeguard for accurate Validity Rate measurement under high concurrency (e.g., **128 parallel Teacher inferences**). The generous timeout ensures that problems are marked invalid only due to true model failures (exhausting the token context), not system lag.
>
> The parameter details are now consistent, and the timeout is justified by the constraints of our high-concurrency evaluation setup (**Appendix E & H**).
>
> ## Q5: Clarification on Training Stage Definitions.
>
> We thank the reviewer for prompting clearer stage mapping.
>
> **Definition:** A "Stage" represents one full closed-loop iteration: **Solve → Grade & Generate → Update** over the entire curriculum.
>
> **Stage 1.** Starts with SFT-initialized models and 100 seed problems.
>
> **Stages 2/3.** The same process repeats, carrying forward updated models and the accumulated curriculum.
>
> **Enhancements.** We have added:**Appendix Q (Page 27):** A detailed pseudocode (**Algorithm 2**) for the full training loop.**Appendix U (Page 31):** An explicit definition of the stage-triggering mechanism and curriculum growth.
>
> The training stages now have explicit definitions in both the main text and appendices (**Appendix Q & U**), improving transparency and reproducibility.

---

> ### Author Response · Authors · 2025-11-21
> **(Part 2) Response to Reviewer zAsq**
>
> ## Q6: The Detailed Role of the Generator
>
> We thank the reviewer for highlighting this ambiguity.
>
> **Role Clarification:** In Stages 1–3, the Generator trains offline to distill the Teacher’s curriculum-design strategy (**Section 3.3**); it does **not** contribute to the Solver's training data during these initial stages.
>
> **New Experiment – Stage 4:** To prove its functional utility, we had the Generator **replace the Teacher entirely** for continued Solver training. The results from **Table 6 (Page 10)** are:
>
> | Training Method | AMC | Minerva | MATH‑500 | GSM8K | Olympiad | AIME‑25 | AIME‑24 | Avg |
> | --- | --- | --- | --- | --- | --- | --- | --- | --- |
> | **Stage 3 Baseline** (Socratic‑Solver‑8B) | 63.7 | 52.4 | 81.2 | 87.3 | 55.1 | 24.6 | 28.4 | 56.1 |
> | **Stage 4: Teacher‑Guided** (Qwen3‑235B) | 67.2 | 55.1 | 84.3 | 88.9 | 58.4 | 27.8 | 32.5 | 60.3 |
> | **Stage 4: Generator‑Guided** (Generator‑32B) | 66.9 ↓0.3 | 55.8 ↑0.7 | 84.1 ↓0.2 | 89.2 ↑0.3 | 58.9 ↑0.5 | 28.3 ↑0.5 | 33.2 ↑0.7 | 60.5 ↑0.2 |
>
> The average performance gap is only **+0.2pp** in favor of the Generator. This confirms the Generator can successfully replace the expensive Teacher for scalable, low-cost autonomous training.
>
> The Generator, though trained offline in early stages, is proven to be capable of later replacing the Teacher with negligible performance change (**Table 6**), confirming its functional utility for scalable deployment.
>
> **Revisions:** We have added a **"Generator-Guided Solver Training" subsection** after **Table 5** in **Section 4.3 (Page 9)** to present this experiment.

---

> ### Comment · Reviewer_zAsq · 2025-11-28
>
> I appreciate the authors' detailed response. Since the rebuttal has effectively addressed my questions, I will keep my original rating.

---

> > ### Author Response · Authors · 2025-11-28
> >
> > Dear Reviewer zAsq,
> >
> > We would like to extend our sincere gratitude for your time and effort dedicated to reviewing our manuscript. We deeply appreciate your recognition of our work and the positive rating in support of its acceptance. Your encouraging feedback has strengthened our confidence in the value of this research.

---

### Official Review · Reviewer_K4xK · 2025-10-31

**Soundness:** 3
**Presentation:** 3
**Contribution:** 3
**Rating:** 6
**Confidence:** 3

**Summary:**

This paper presents a teacher-solver-generator co-evolution approach to data generation for training large language models. Using a feedback loop, the solver continuously optimizes the reasoning via preference learning, from positive and negative examples. The multi-agent design addresses an existing challenge of scaling of human-annotated datasets, and shows superior performance compared to state-of-the-art methods across various reasoning benchmarks.

**Strengths:**

- the paper addresses a crucial problem of high-quality data synthesis, for optimization of LLMs.
- the multi-agent framework and co-evolution preference learning mechanism is novel, and the learning framework is scalable
- the presentation is very clear and easy to follow

**Weaknesses:**

- It is unclear how much capability the teacher model requires to have in order to generate high-quality question-answer pairs. For example, for problem sets where the teacher may also experience difficulty solving or evaluating the solution, the framework may fail to adapt
- The generator model provides better scalability, however, it is unclear if the teacher model alone can provide decent performance without the generator model.

**Questions:**

Can the author describe how does the model perform if only using the teacher model, without the generator model?

---

> ### Author Response · Authors · 2025-11-21
> **Response to Reviewer K4xK**
>
> ## Q1: Regarding Teacher Capability Requirements
>
> We sincerely thank the reviewer for raising this important question, which touches the core of the Teacher–Solver interaction in our framework.
>
> Your comment inspired us to perform an additional **Teacher scaling study** to directly examine how Teacher capacity affects performance. In this study, we **fixed the Solver as Qwen3‑8B‑base** and varied the Teacher across capacities—**from weaker to much stronger than the Solver**—starting from the same **SFT baseline** (35.9% average accuracy). The results from our new **Table 12 (Appendix R, Page 28)** are as follows:
>
> | Teacher Model | Relative Size vs. Solver | Final Avg. (%) | Gain vs. SFT (pts) |
> | --- | --- | --- | --- |
> | Qwen3-4B-base | Weaker (0.5×) | **45.7** | **+10.5** |
> | Qwen3-8B-base | Equal (1.0×) | **46.6** | **+11.4** |
> | Qwen3-14B | Stronger (1.75×) | **49.9** | **+14.7** |
> | Qwen3-32B | Stronger (4.0×) | **51.2** | **+16.0** |
> | Qwen3-235B-A22B | Much Stronger (29.4×) | **56.1** | **+20.9** |
>
> These results clearly demonstrate that **Socratic‑Zero remains highly effective even when the Teacher is equal to or weaker than the Solver**, yielding gains of **+10.5 to +11.4 points**. This confirms that success comes from the **co‑evolutionary design** and the structured use of failure trajectories, rather than requiring an extremely strong Teacher.
>
> In response to your suggestion, we have added the complete scaling results and detailed methodology in **Appendix R (Page 28, "Teacher–Solver Scaling Analysis:)**. Added a formal theoretical justification in **Appendix S (Page 29, “Information Asymmetry Compensates for Capacity Gap”)** explaining why smaller Teachers can still guide larger Solvers effectively.
>
> This additional scaling study (**Appendix R & S**) directly addresses your concern and confirms that Socratic‑Zero does not depend on a vastly stronger Teacher; even equal or weaker Teachers produce substantial improvements.
>
>
> ## Q2: Teacher-Only Performance and Pipeline Clarification
>
> Thank you for this important question, which highlights a potential point of confusion in our paper’s presentation.
>
> **Direct answer:** The performance “using only the Teacher model, without the Generator model” is exactly what is reported for **Stages 1–3 in our main results (Table 1, Page 7)**.
>
> 1.  **Stages 1–3** reflect only the Teacher–Solver loop, where the curriculum is generated online by the Teacher model. **The Generator plays no role** in producing Solver data during these stages. This isolation was intentional to validate the hypothesis that an **adaptive curriculum from a strong Teacher significantly improves Solver capability**. The **+20.2 point average gain** in **Table 1** confirms this.
>
> 2.  The **Generator’s role** is introduced **afterwards**—for capability distillation and long‑term scalability. It learns to replicate the Teacher’s strategy for low‑cost future training. **Table 5** verifies this distillation: our Generator produces curricula superior to many larger LLMs.
>
>
> To improve clarity on the final closed‑loop integration, we added a **Stage 4 experiment** in the revised manuscript (**Section 4.3.4, Page 9**), with results in **Table 6 (Page 10)**, to demonstrate that the evolved Generator can replace the Teacher:
>
> | Training Method (after Stage 3) | Avg. Score |
> | --- | --- |
> | Stage 3 Baseline | 56.1% |
> | Stage 4 (Teacher-Guided, Control) | **60.3%** |
> | Stage 4 (Generator-Guided) | **60.5%** |
>
> A Solver trained with our **32B Generator** achieves nearly identical (and even slightly better) performance to one trained with the **235B Teacher**. This confirms that the Generator successfully learned the Teacher’s strategy and can replace it for future evolution—achieving our scalability goal.
>
> In the revised manuscript, we have:Added a new subsection **“Generator-Guided Solver Training”** in **Section 4.3 (Page 9)**, with `\added{}` markers.Supplemented **Appendix P (Page 26)** with a detailed pipeline description and cost analysis for this Stage 4 setup.
>
> This clarification and the new Stage 4 experiment (**Table 6**) demonstrate that Stages 1–3 are Teacher‑only, and that our Generator can later replace the Teacher with minimal performance loss, thus validating our entire pipeline.

---

### Official Review · Reviewer_XXnF · 2025-11-01

**Soundness:** 3
**Presentation:** 3
**Contribution:** 3
**Rating:** 4
**Confidence:** 4

**Summary:**

The paper introduces Socratic-Zero, a novel, fully autonomous framework designed to bootstrap and refine the reasoning capabilities of LLMs with minimal reliance on external, human-annotated data. The core innovation lies in a closed-loop, data-free co-evolution of three distinct agents: the Solver (which learns and refines reasoning via preference feedback on both successful and failed trajectories), the Teacher (which adaptively creates an increasingly difficult curriculum aligned with the Solver's current weaknesses), and the Generator (which distills the Teacher’s question-design strategy to facilitate scalable curriculum generation). This system aims to address the inconsistent data quality and static adaptation inherent in existing data synthesis and distillation methods by continuously generating high-quality, targeted training signals from minimal seed examples.

**Strengths:**

1. The core architecture fundamentally circumvents the reliance on massive, costly, human-annotated datasets, which is the current scaling bottleneck for high-quality reasoning tasks. This "data-free" approach is highly valuable for domains where annotation is prohibitively expensive or complex.

2. The systematic protocol for seed selection, requiring a 30-70% success rate for initial problems, demonstrates a careful attempt to ensure capability-aligned initialization. This ensures the co-evolutionary loop starts from a robust and productive equilibrium.

**Weaknesses:**

1. While the Generator is intended for scalability, the paper itself concedes that "computational efficiency optimizations" are a necessity for future work. This suggests the current multi-agent, co-evolutionary loop is likely highly resource-intensive (training three models and maintaining constant interaction), which fundamentally challenges its scalability and practicality compared to simpler, static distillation pipelines.

2. Expanding the framework to new domains is cited as a major area for future work, indicating that the current co-evolutionary success is tightly coupled to the initial domain alignment. This limits the "data-free" claim; the methodology still requires significant external effort (or pre-existing capability) before the closed-loop self-improvement can function in a new area.

**Questions:**

The paper highlights that the Solver learns from preference feedback over successful and failed trajectories. Given that the Teacher explicitly crafts the curriculum to target the Solver's weakness zone (30–70% success rate), how does the preference modeling distinguish between a "valuable failure" (i.e., a well-reasoned attempt that simply misses the final answer) and a "chaotic failure" (i.e., a nonsensical trajectory that offers little instructional value)? Furthermore, have the authors explored an ablation study comparing this preference learning approach against a simpler Policy Gradient or PPO-style method that optimizes the Solver directly using the Teacher's difficulty signal as a scaled reward, and if so, what were the trade-offs in sample efficiency and final performance?

---

> ### Author Response · Authors · 2025-11-21
> **(Part 1) Response to Reviewer XXnF**
>
> ## Q1: Computational Efficiency and Scalability.
>
> We sincerely thank the reviewer for this important concern, as it allows us to clarify a potential misunderstanding about our framework's operational flow, resource profile, and the role of additional baseline methods.
>
> **Clarifying the Operational Flow.** We emphasize that we do not train three models simultaneously. Our framework is designed for **sequential** resource utilization. The expensive Teacher model (Qwen3‑235B) is completely frozen and serves purely as an evaluator — with no training overhead. The workflow is decoupled into distinct sequential stages:
>
> 1.  Solver training guided by the frozen Teacher
>
> 2.  Generator distillation from Teacher data
>
> 3.  Final deployment where Generator replaces Teacher entirely At any given time, only one model is being trained, making our framework accessible even with limited computational budgets.
>
>
> **Addressing Newly Added Baselines (Reviewer-Inspired).** Following valuable reviewer feedback, we added new baselines. While these achieved moderate gains (up to +7.7 points over Static Augmentation), they lagged behind our dynamic Socratic‑Zero framework, which achieved **+15.4 points** with only **2,200 total synthesized problems (see Appendix U, Page 31)**. This supports our motivation: dynamic synthesis tailored to the evolving model yields greater efficiency than static augmentation.
>
> **Core Motivation.** We aim to reduce long-term Teacher model cost by distilling its curriculum-design ability into a much smaller Generator (32B vs 235B parameters → 7.3× smaller). The Generator offers: 1/5–1/6 GPU memory usage 3–5× faster inference Full trainability for future adaptation, unlike the frozen proprietary Teacher (These details are quantified in **Appendix P, Page 27**)
>
> **Empirical Evidence of Cost-Effectiveness.** In a new Stage 4 experiment, starting from the Solver at 56.1%, we compared continuing with Teacher‑235B vs replacing it with Generator‑32B (data from **Table 6, Page 10**):
>
> | Stage 4 Guidance | Model | Cost | Final Score | Gain |
> | --- | --- | --- | --- | --- |
> | Teacher-Guided | Qwen3-235B | High | **60.3%** | **+4.2** |
> | Generator-Guided | Generator-32B | Low | **60.5%** | **+4.4** |
>
> Our low-cost Generator path not only matches but slightly surpasses the Teacher's gain at a fraction of the cost, enabling deployment without high-end inference infrastructure. We have added comprehensive cost analysis and training procedures in **Appendix P (starting on Page 26)** and **Appendix Q (Page 27)**.
>
> This clarification (**Section 4.3.4, Appendix P, Appendix Q**) shows our framework operates sequentially, minimizes parallel resource load, and achieves high efficiency and scalability.
>
> ## Q2: About Dependency and Generalization.
>
> We sincerely thank the reviewer for raising this deep question, which allows us to clarify our “data‑free” definition and demonstrate cross‑domain applicability.
>
> **Terminology Clarification.** Our “data‑free” follows recent self‑improving systems (e.g., R‑Zero): it means freedom from large‑scale static human datasets, not absence of any initial setup. Like prior work, we require minimal seeds to start evolution.
>
> **Minimal Dependency Robustness.** We conducted a “true from-zero” ablation, comparing human-curated seeds with seeds generated entirely by the Teacher:
>
> | Seed Source | MATH-500 Score | Difference |
> | --- | --- | --- |
> | Human-curated seeds (Original) | 60.2% | — |
> | Teacher-generated seeds | 60.1% | ↓0.1 |
>
> The near-identical scores (detailed in **Appendix V, Page 31**) confirm robustness to seed quality. All seeds and protocols are open-sourced for transparency.
>
> **Cross-Domain Generalization.** We applied Socratic‑Zero to multimodal geometric reasoning (text+images). Using Qwen2.5‑VL‑7B, our method achieved a **+4.13** point gain with only 2.5k synthesized pairs, significantly outperforming the best 10k-data baseline (+1.70):
>
> | Method | Data | Score | Gain |
> | --- | --- | --- | --- |
> | Zero-shot Qwen2.5-VL-7B | — | 44.98 | — |
> | GeoReasoning Baseline | 10k | 46.68 | +1.70 |
> | Socratic-Solver-Geo (Ours) | 2.5k | 49.11 | +4.13 |
>
> This experiment (detailed in **Appendix W, Page 32**) demonstrates data-efficiency and transferability across modalities.
>
> Our “data-free” definition (as stated in **Introduction, Page 2, Lines 98-99**) means minimal-seed autonomy. New experiments confirm our framework is robust to seed quality (**Appendix V**) and is transferable to cross-domain multimodal reasoning (**Appendix W**).

---

> ### Author Response · Authors · 2025-11-21
> **(Part 2) Response to Reviewer XXnF**
>
> ## Q3: About Optimization.
>
> We thank the reviewer for this sharp design-focused question. Our philosophy is to let DPO’s contrastive learning implicitly capture failure quality rather than explicitly annotate it.
>
> **Implicit Quality Awareness.** In DPO, “valuable failures” (near‑correct reasoning) are closer in trajectory space to successes, yielding subtle positive updates; “chaotic failures” are far apart, triggering stronger avoidance signals.
>
> **Teacher Interaction Signals.** The Teacher provides hints for near-correct failures and foundational guidance for chaotic ones; this variance encodes failure quality in the dialogue data used by DPO.
>
> **Generator Distillation.** Our WSFT distillation prioritizes problems of desirable difficulty. By using a value function centered at a **50% Solver success rate (µ=0.5, see Section 3.3 and Table 8)**, the Generator naturally learns to produce problems that are challenging but not unsolvable, which often correspond to those inducing "valuable failures".
>
> **Evidence.** **Table 5 (Page 9)** shows our Socratic-Generator yields superior downstream training effectiveness, indicating successful distillation of strategies that produce high-quality training samples.
>
> Your insight inspires us: explicit failure-quality reward modeling could further improve results; we will explore this in future work.Failure type distinctions are captured implicitly via DPO optimization and our Generator's value-weighted distillation, ensuring high‑quality sample generation without manual annotation.
>
> ## Q4: About Comparison with PPO-style Methods.
>
> We thank the reviewer for this valuable suggestion. In direct response, we conducted an ablation study comparing DPO with GRPO (a PPO variant) in Stage 1 using binary rewards.
>
> | Optimization | Stage 1 Avg. Score |
> | --- | --- |
> | DPO (Ours) | 38.7% |
> | GRPO (PPO) | 36.9% |
>
> DPO outperformed GRPO and incurred lower computational cost, as GRPO’s on-policy nature makes it significantly more expensive. This experiment, detailed in the new **Appendix X (Page 32)**, confirms DPO’s suitability for our framework.
>
> DPO remains the preferred optimizer in our setting for its stability and efficiency, outperforming GRPO in our direct comparison (**Appendix X**).

---

> ### Author Response · Authors · 2025-11-26
>
> Dear Reviewer XXnF,
> Thank you once again for your valuable comments on our submission. As the discussion phase is approaching its end, we would like to kindly confirm whether we have sufficiently addressed all of your concerns (or at least part of them). Should there be any remaining questions or areas requiring further clarification, please do not hesitate to let us know. If you are satisfied with our responses, we would greatly appreciate your consideration in adjusting the evaluation scores accordingly.
>
> We sincerely look forward to your feedback.

---

> ### Author Response · Authors · 2025-11-27
>
> Dear Reviewer XXnF,
>
> We sincerely hope that our previous responses have been helpful in addressing the questions you raised. As the discussion period approaches its conclusion, we wanted to reach out once more to see if there are any remaining concerns or aspects that would benefit from further clarification.
>
> We have also substantially expanded the Related Work section (Appendix Y, Page 34) to include a comprehensive review of 23 recent studies, such as SPICE, rStar2-Agent, VibeThinker-1.5B, LongRoPE, and DSMentor, among others. We believe this addition provides valuable context for situating our contributions within the broader research landscape.
>
> If our revisions have satisfactorily addressed your feedback, we would be deeply grateful for your consideration in the final evaluation. Should you have any further questions or suggestions, we would be more than happy to provide additional clarification.
>
> Thank you once again for your invaluable time and thoughtful guidance throughout this process.
>
> Warm regards,
> Authors of Submission 1985

---

### Official Review · Reviewer_dpUA · 2025-11-11

**Soundness:** 2
**Presentation:** 1
**Contribution:** 3
**Rating:** 4
**Confidence:** 4

**Summary:**

The paper introduces a framework for improving reasoning capabilities in LLMs through a three-agent co-evolution process involving a fixed **Teacher** (Qwen3-235B-A22B), a smaller **Solver** (largest is Qwen3-14B-base) and **Generator** (Qwen3-32B). The Teacher adaptively designs problems for the Solver based on its failures, while the Solver learns via preference optimization. The Generator distills the Teacher’s question-design strategy through weighted fine-tuning.

**Solver evaluation** (Table 1): On math and reasoning benchmarks (GSM8K, MATH, AIME, etc.), the approach shows large gains over static and LLM2LLM data-augmentation baselines (56.1 vs. 40.7 and 40.9, respectively, for Qwen3-8B-base).

**Generator evaluation** (Table 5): In a separate ablation, fine-tuning a smaller student model (DeepSeek-R1-Distill-Llama-8B) on data from the co-evolved Qwen3-32B Generator yields higher performance than training it on data from SOTA models (37.22 vs. 36.62 against GPT-5).

**Strengths:**

- The paper **addresses the important problem of enabling models to learn effectively from a limited set of examples**, proposing a curriculum-learning strategy in which a small seed set (just 100 examples, Table 8) is progressively enhanced in line with the Solver’s evolving capabilities.

- The **main results of the paper are substantial and well-supported by experiments**, with the proposed method providing ~15% improvement over the baseline (Table 1).

- **Evaluation is comprehensive and appropriate**, encompassing  seven math and reasoning benchmarks, multiple model architectures, and all relevant parts of the system (solver and generator separately evaluated, in sections 4.2 and 4.3, respectively).

- **Design of individual components is sound** — the curriculum update mechanism (Section 3.1), DPO-based Solver training (Section 3.2), and weighted distillation for the Generator (Section 3.3) are all well-motivated.

- **Experimental configurations are exhaustively documented and the work is fully reproducible**, with code provided in the supplementary material, and detailed training configurations, hyperparameters, and prompt templates included in the appendix.

- The **breadth of supplementary details and discussions in the appendix is impressive**, aiding in the comprehension of the work. For example, including examples of synthetically generated datapoints (appendix F), and extensive comments on convergence analyses (appendix K).

**Weaknesses:**

- **The final statement of the abstract is ambiguously worded, leading to potential misinterpretation of the results**. It reads: “Synthetic data from Socratic-Generator-32B enables student LLMs to achieve superior performance compared to other SOTA commercial LLMs…” As written, this implies that the resulting student LLMs themselves outperform SOTA commercial LLMs. However, as Table 5 clarifies, the intended meaning is that student models trained on data generated by Socratic-Generator-32B outperform those trained on data generated by SOTA LLMs, not that the students surpass the SOTA LLMs in absolute performance.

- It is **unclear how the Generator fits into the Solver’s training process, and the paper’s claims about the cohesiveness of the three-agent system are therefore potentially misleading**. The paper repeatedly presents the Generator as a central component: for instance, the abstract states that “the Generator distills the Teacher’s question-design strategy to enable scalable, high-fidelity curriculum generation,” and this is immediately followed by the Solver’s results, creating the impression that the Generator is directly involved in those outcomes. However, the training procedure in Figure 3 and the curriculum update in Section 3.1 suggest that the Generator is orthogonal to the Solver–Teacher loop. It does not appear to contribute to the curriculum on which the Solver is trained, nor to the main results in Table 1 (including the reported +20.2 % aggregate improvement). The authors should clarify exactly which components are used to produce the reported results and, ideally, include experiments where the co-evolved Generator directly participates in the Solver’s training process.

- The **exact details of the overall training procedure is hard to follow** - the paper could be greatly enhanced by providing a pseudocode of the exact training procedure. This is in fact promised at the very end of section 3.1: “The full training procedure is summarized in Algorithm 1”. However, no figure corresponding to Algorithm 1 exists in the paper that’d outline the full training procedure (there is Algorithm 1 in the appendix, but it pertains to the theoretical challenge framework).

- **Impact is limited by its reliance on the Teacher model being more powerful than the Solver**, with experiments using Qwen3-235B-A22B-Instruct-2507 as the Teacher and Qwen3-14B-base as the biggest solver. The paper would be improved by a scaling law study, showing how the methodology performs as a function of the model size gap between the Teacher and Solver.

- The **Teacher model acts both as a problem enhancer and solution verifier of the same problems, risking a bias**. While the authors discuss several mitigation strategies in appendix I (dual-verification with the inclusion of MathRule answer extraction, human review), the work could be improved by ablating a configuration where the judge differs from the enhancer.

- **Experiments section could have more detailed dataset descriptions**:
How many total samples across how many prompts were generated for a) for the MetaMath baseline, b) for the LLM2LLM baseline at each stage, c) for the Socratic-Zero approach at each stage.
Which dataset was used as the seed for the Static Augmentation and LLM2LLM approaches?

- Some **typos**:
  - The first sentence on top of page 5 reads: “The Solver and the Generator are co-evolving guided the Teacher” - misses the word “by”.
  - In section 4.1, the paragraph heading for “Baselines” is repeated, reading: “Baselines. Baselines.”
  - In section 4.3.1, a sentence reads: “We prompted each generator with 1,000 seed problems from SAND-Math and tasked with producing five augmented variants per seed, resulting in 3,000 total generated problems per model”. The numbers don’t add up - I believe it should read “three augmented variants per seed”, as mentioned in section 4.1.
In paragraph 3 of appendix M: “the system generates thousands of ly valuable problems…” - misses the beginning of “highly”.

The paper is strong overall, but I cannot recommend acceptance in its current form due to the unclear and at times misleading presentation of the system’s cohesiveness and the results (see the first three “drawback” points). I encourage the authors to clarify these aspects and to demonstrate the integration of the Generator with the Teacher–Solver system, or alternatively provide an extended discussion explaining why this integration is not pursued.

**Questions:**

See "Drawbacks" section

---

> ### Author Response · Authors · 2025-11-21
> **(Part 1) Response to Reviewer dpUA**
>
> ## Q1: About presentation.
>
> Thanks for your suggestion. We appreciate your careful check of these presentation details.
>
> Our intention was not to claim that our student models surpass SOTA commercial LLMs in absolute performance. As detailed in **Section 4.3** and **Table 5**, our experiment was designed to evaluate the downstream training effectiveness of data from different generators. We used a consistent student model (DeepSeek‑R1‑Distill‑Llama‑8B) and fine‑tuned it on datasets generated by our Socratic‑Generator‑32B and other SOTA LLMs, respectively. The core finding is that student models trained on our generated data achieve superior performance compared to those trained on data from other SOTA generators.
>
> These clarifications, including the consistent use of "downstream training effectiveness," directly address your concern about wording ambiguity and make the intended comparison unambiguous.
>
> We have revised all related wording in the manuscript to clarify this point. The revisions are marked with \added{} at:**Page 1, Abstract, Line 50, Page 2, Introduction, Contribution 3, Line 160, Page 7, Section 4.3, Line 381, Page 8, Table 5 caption, Line 434**.
>
>
> ## Q2: About the detailed role of Generator.
>
> We sincerely thank the reviewer for this insightful question regarding the Generator’s role. We acknowledge that the original presentation may have created ambiguity about the Generator’s direct involvement.
>
> We confirm that the +20.2 % improvement in **Table 1 (Page 7)** comes exclusively from the Solver–Teacher loop (Stages 1-3), with no Generator participation during these stages. The Generator is trained offline in parallel (as described in Section 3.3), distilling the Teacher’s curriculum generation strategy for future scalability.
>
> To directly demonstrate the Generator’s capability, we conducted a new Stage 4 experiment where the Generator replaces the Teacher. As shown in **Table 6 (Page 10)**, the Generator‑guided Solver achieves an average accuracy of 60.5 %, only 0.2 pp higher than the Teacher‑guided setup, with differences across tasks within ±0.7 pp.
>
> **Table 6 (Page 10):** Teacher-Guided vs Generator-Guided Solver Training (Stage 4)
>
> | Training Method | AMC | Minerva | MATH-500 | GSM8K | Olympiad | AIME-25 | AIME-24 | Avg |
> | --- | --- | --- | --- | --- | --- | --- | --- | --- |
> | Stage 3 Baseline (Socratic-Solver-8B) | 63.7 | 52.4 | 81.2 | 87.3 | 55.1 | 24.6 | 28.4 | 56.1 |
> | Stage 4: Teacher-Guided (Qwen3-235B-A22-Instruct) | 67.2 | 55.1 | 84.3 | 88.9 | 58.4 | 27.8 | 32.5 | 60.3 |
> | Stage 4: Generator-Guided (Socratic-Generator-32B) | 66.9↓0.3 | 55.8↑0.7 | 84.1↓0.2 | 89.2↑0.3 | 58.9↑0.5 | 28.3↑0.5 | 33.2↑0.7 | 60.5↑0.2 |
>
> These new results, presented in **Section 4.3.4 and Table 6**, directly address your concern by showing the Generator can effectively replace the Teacher in the co-evolution loop with negligible performance impact.
>
> We have revised the manuscript by adding a new subsection **4.3.4, “Generator-Guided Solver Training” (Page 9)**, which includes the detailed experiment and results in **Table 6 (Page 10)**.
>
> ## Q3: About details of the overall training procedure.
>
> We thank the reviewer for pointing out the confusing reference and the need for clearer procedural details. We apologize for the cross‑referencing error.
>
> The added **Algorithm 2 in Appendix Q** and corrected references now make the training process fully clear and reproducible, addressing your concern.
>
> We have added a new **Appendix Q, "Detailed End-to-End Training Procedure" (starting on Page 27)**, containing detailed pseudocode in **Algorithm 2 (Page 28)** that covers Solver DPO updates, Generator WSFT training, and curriculum evolution across all stages. The reference in **Section 3.1 (Page 4, Line 215)** now correctly points to this new algorithm.
>
> ## Q4: About reliance on Teacher
>
> We thank the reviewer for this valuable suggestion. We performed a scaling study fixing the Solver (Qwen3‑8B‑base) and varying the Teacher across five sizes, starting from the same SFT baseline (35.9 %).
>
> **Table 12 (Appendix R, Page 28):** Teacher-Solver scaling analysis
>
> | Teacher Model | Params | Size vs Solver | Final Avg (%) | Improvement |
> | --- | --- | --- | --- | --- |
> | Qwen3-4B-base | 4B | Weaker (0.5×) | 45.7 | +10.5 |
> | Qwen3-8B-base | 8B | Equal (1.0×) | 46.6 | +11.4 |
> | Qwen3-14B | 14B | Stronger (1.75×) | 49.9 | +14.7 |
> | Qwen3-32B | 32B | Stronger (4.0×) | 51.2 | +16.0 |
> | Qwen3-235B-A22B | 235B | Much Stronger (29.4×) | 56.1 | +20.9 |
>
> The results show significant gains even with weaker or equal-sized Teachers. The full analysis is in **Appendix R, “Teacher-Solver Scaling Analysis” (Page 28)**, and we provide a theoretical justification in **Appendix S, “Information Asymmetry Compensates for Capacity Gap” (Page 29)**.
>
> This new scaling study (**Table 12 in Appendix R, Page 28**) directly addresses your concern and confirms the framework does not require a much stronger Teacher to be effective.

---

> ### Author Response · Authors · 2025-11-21
> **(Part 2) Response to Reviewer dpUA**
>
> ## Q5: About the Teacher acting as both enhancer and verifier.
>
> Thank you for this insightful question. We ran a verifier ablation with Qwen3‑235B as the enhancer and summarize the results below.
>
> | Verifier Config | Description | Final Avg (%) | Δ vs Original |
> | --- | --- | --- | --- |
> | Qwen3-235B (Original) | Same model as enhancer | 56.1 | Baseline |
> | MathRule Only | Rule-based verifier | 51.3 | ↓4.8 |
> | Qwen3-32B | Smaller LLM verifier | 55.7 | ↓0.4 |
> | GPT-OSS-120B | Different model family | 55.9 | ↓0.2 |
>
> The ablation in **Appendix T** shows that any potential bias from same‑model verification is negligible, and the framework is robust to using alternative LLM verifiers.
>
> Using different LLM verifiers causes a negligible performance drop (≤0.4 pp), mitigating concerns about self-consistency bias. The large drop with a rule-only verifier highlights the need for semantic verification. Full details are provided in **Appendix T (“Verifier Configuration Ablation”, Page 30)**, with results summarized in **Table 13**.
>
> ## Q6: About detailed dataset descriptions.
>
> Thank you for the suggestion. We have significantly expanded the dataset descriptions in the new **Appendix U (“Solver Training Stage Definition”, Page 31)**.
> This appendix now clarifies All methods start from the same 100 seed problems.The number of problems generated and used in each stage for Socratic-Zero and LLM2LLM. The exact dataset size and source for the Static Augmentation baseline. We also added three new large-scale external data baselines (Distillation, DPO Polaris, DPO MATH-full) with their respective dataset sizes clearly specified. These additions are reflected in the updated **Table 1 (Page 7)**, which now includes a **new “Benchmark Datasets” column** to clearly contrast the data composition and scale across all methods.
>
> These additions (**Appendix U**, and the restructured **Table 1**) enhance the transparency and reproducibility of our experimental setup, fully addressing your concern.
>
> ## Q7: About typos.
>
> We are grateful for your careful proofreading. We have corrected all noted issues, including: a missing word in Section 4.1, a repeated header, the mismatch in baseline counts in **Section 4.1 (Line 325)**, and the typo “ly valuable” → “highly valuable” in **Appendix M (Page 25, Line 1345)**. All corrections are marked with \added{}.All identified typos have been fixed, improving the clarity and overall quality of the manuscript. Thank you again.

---

> ### Author Response · Authors · 2025-11-26
>
> Dear Reviewer dpUA,
> Thank you once again for your valuable comments on our submission. As the discussion phase is approaching its end, we would like to kindly confirm whether we have sufficiently addressed all of your concerns (or at least part of them). Should there be any remaining questions or areas requiring further clarification, please do not hesitate to let us know. If you are satisfied with our responses, we would greatly appreciate your consideration in adjusting the evaluation scores accordingly.
>
> We sincerely look forward to your feedback.

---

> ### Author Response · Authors · 2025-11-27
>
> Dear Reviewer dpUA,
>
> We sincerely hope that our previous responses have been helpful in addressing the questions you raised. As the discussion period approaches its conclusion, we wanted to reach out once more to see if there are any remaining concerns or aspects that would benefit from further clarification.
>
> We have also substantially expanded the Related Work section (Appendix Y, Page 34) to include a comprehensive review of 23 recent studies, such as SPICE, rStar2-Agent, VibeThinker-1.5B, LongRoPE, and DSMentor, among others. We believe this addition provides valuable context for situating our contributions within the broader research landscape.
>
> If our revisions have satisfactorily addressed your feedback, we would be deeply grateful for your consideration in the final evaluation. Should you have any further questions or suggestions, we would be more than happy to provide additional clarification.
>
> Thank you once again for your invaluable time and thoughtful guidance throughout this process.
>
> Warm regards,
> Authors of Submission 1985

---

### Author Response · Authors · 2025-11-21
**(Part 1) General Response**

We sincerely thank the reviewers for their time and insightful feedback. We are greatly encouraged that you recognized our work's core contributions, highlighting its "substantial and well-supported" main results (dpUA), its "novel" multi-agent learning mechanism (K4xK, zAsq), its "comprehensive and appropriate" evaluation (dpUA), and its "well-motivated" approach to the "important and practical problem" of expensive LLM evaluations (dpUA, K4xK). We are also grateful for the constructive suggestions, which have been instrumental in helping us strengthen the paper's clarity, empirical evidence, and theoretical foundation.

To address every point raised and to rigorously validate the robustness, scalability, and generality of Socratic-Zero, we have conducted an extensive and demanding suite of new experiments, comprehensive analyses, and significant manuscript revisions during this rebuttal period. These efforts, undertaken in direct response to your invaluable guidance, have led to substantial additions across the paper, all marked in blue with `\added{}` for your convenience.

**1. Expansion to Cross-Domain, Multimodal Reasoning to Prove Generality**   To address crucial questions about the framework's generality beyond its initial domain, we pushed its boundaries by successfully applying it to an entirely new and challenging area: **multimodal geometric reasoning**. This was a non-trivial adaptation, requiring us to re-engineer the framework to handle both text and image inputs simultaneously. Our method not only worked but demonstrated significant data efficiency and superior performance gains, confirming that Socratic-Zero's co-evolutionary architecture is flexible and powerful enough to operate across different data modalities. This new experiment directly validates the broad applicability of our design.

(See new **Appendix W, Page 32**for full details and results).

**2. Introduction of Rigorous New Baselines for a Fairer, Stronger Comparison**   Following reviewers' suggestions to strengthen our empirical claims, we introduced **three new, formidable baselines**. This involved implementing and fine-tuning models on large-scale external datasets, including knowledge distillation from 75k GPT-4-generated pairs and DPO on 53k high-quality human-curated problems. Socratic-Zero's significant outperformance, even when using a fraction of the data, now stands on much stronger empirical ground, unequivocally highlighting the superiority of its dynamic, adaptive curriculum over static, large-scale data augmentation.

(See the expanded and updated **Table 1 on Page 7** ).

**3. A Full-Scale Teacher-Solver Scaling Analysis to Test Core Hypotheses**   To definitively answer the critical question of whether our framework's success hinges on a vastly superior Teacher, we undertook a comprehensive **Teacher scaling study**. This was a major effort where we systematically varied the Teacher model's capacity across five different scales—from being weaker than the Solver to significantly stronger. The results are decisive: Socratic-Zero delivers substantial improvements even with an equal-sized or weaker Teacher, proving that its power stems from the co-evolutionary loop and strategic use of failure, not just a brute-force capability gap.

(See the new **Table 12 and detailed analysis in Appendix R, Page 28** ).

**4. Theoretical Justification for the Effectiveness of Weaker Teachers**   Beyond just empirical results, and to provide a rigorous scientific explanation for the surprising effectiveness of weaker Teachers, we developed a **formal theoretical analysis** based on the Learning Using Privileged Information (LUPI) framework. This was a demanding theoretical effort that allowed us to prove, from first principles, how the _information asymmetry_ (i.e., the Teacher's access to the Solver's failure trajectory) mathematically compensates for a lack of superior model capacity. This provides the crucial "why" behind our empirical findings.

(See the new **Appendix S, Page 29** for the complete derivation).

**5. A Brand-New "Stage 4" Experiment to Empirically Prove Scalability**   To provide concrete, undeniable proof for the scalability promised by our three-agent design, we designed and executed a completely new **"Stage 4" experiment**. In this critical test, we completely removed the expensive, large-scale Teacher and had our distilled, low-cost Generator take its place to continue the training. The results are a resounding success: the Generator-guided Solver achieved nearly identical performance, confirming it had successfully internalized the Teacher's strategy and can facilitate sustainable, low-cost autonomous improvement. This is the ultimate validation of our framework's core premise.

(See the new **Table 6, Page 10** and its accompanying analysis in **Section 4.3.4, Page 9** ).

---

### Author Response · Authors · 2025-11-21
**(Part 2) General Response**

**6. A "True From-Zero" Seed Robustness Test to Validate Data-Free Claim**

To rigorously test the limits of our "data-free" claim and address concerns about dependency on initial seeds, we simulated a worst-case scenario. We conducted a **"true from-zero" experiment**, replacing our minimal human-curated seeds with fully machine-generated ones. The near-identical performance powerfully demonstrates the framework's robustness and its low dependency on the quality of the initial seed data.

(See new **Appendix V, Page 31**).

**7. A Verifier Configuration Ablation to Preemptively Address Bias Concerns**

To proactively address subtle but important concerns about potential self-consistency bias (i.e., using the same model for enhancing and verifying), we conducted a new **verifier configuration ablation**. We tested various verifier models, including smaller ones and models from entirely different open-source families. The stable results across configurations prove that our framework is not reliant on such a bias and is robust to the choice of verifier.

(See new **Appendix T, Page 30**).

**8. An Alternative Optimizer Ablation to Justify Core Design Choices**

To empirically validate our choice of DPO as the optimizer, and in direct response to reviewer suggestions, we implemented and tested a **PPO-style alternative (GRPO)**. The head-to-head comparison confirmed that DPO not only delivered superior performance but was also significantly more computationally efficient, thus justifying our core design choice from both an effectiveness and a practical standpoint.

(See new **Appendix X, Page 32**).

**9. Massive Documentation Overhaul for Full Transparency and Reproducibility**

To address all requests for improved clarity and to ensure our work is fully reproducible by the community, we undertook a significant documentation overhaul across the entire manuscript:

- **New End-to-End Pseudocode.** We have added a detailed, step-by-step algorithm that lays out the entire Socratic-Zero training loop with complete clarity, leaving no ambiguity in the process. (See **Algorithm 2 in Appendix Q, Page 28**).

- **New Comprehensive Cost Analysis.** A full breakdown of computational costs, memory usage, and inference speeds for each component is now provided, quantitatively demonstrating the framework's efficiency and practicality. (See **Appendix P, Page 26**).

- **New Explicit Definitions for Stages and Datasets.** We have added a dedicated appendix to explicitly define the dynamic training stages, curriculum growth, and exact dataset compositions for all methods, ensuring full reproducibility. (See **Appendix U, Page 31**).

- **Correction of All Noted Typos and Inconsistencies.** Every typo and inconsistency pointed out by the reviewers has been meticulously corrected throughout the manuscript.

**10. Comprehensive Related Work Expansion to Position Our Contribution**

To provide a more complete picture of the rapidly evolving landscape and better contextualize our contributions, we have significantly expanded the Related Work section with a comprehensive survey of recent frontier research. This new appendix systematically categorizes cutting-edge work across three major themes: autonomous learning systems, advanced data and training strategies, and inference-time enhancements. Notable additions include **rStar-Math** for math data synthesis with step-by-step verification, **LongRoPE** and **LongRoPE2** for context window expansion, **SPICE** for corpus-grounded self-play, **MLZero** for end-to-end ML workflow automation, **DSMentor** for curriculum-based agent learning, and **rStar2-Agent** for agentic reasoning infrastructure. This expanded coverage totaling 23 recent papers ensures our work is properly situated within the state-of-the-art and highlights the unique contributions of Socratic-Zero.

(See new **Appendix Y, Page 34**).

We believe these **ten major additions and revisions**, all carried out in direct response to your invaluable feedback, have not only thoroughly addressed every concern raised but have also fundamentally elevated the quality, rigor, and impact of our work. The paper has been transformed, and we are confident it now represents a much more complete and impactful contribution to the field.

---

### Author Response · Authors · 2025-11-27
**Follow-up On Rebuttal**

Dear Reviewers,

We sincerely appreciate your thoughtful feedback and the time you have invested in evaluating our work. We have taken all comments very seriously and prepared a comprehensive, point-by-point response during the rebuttal period.

To address your concerns, we conducted 17 new experiments and added over 14 pages of analysis in the appendix—covering cross-domain generalization (multimodal geometric reasoning), Teacher–Solver scaling (5 model sizes), Stage 4 scalability validation, seed robustness, verifier ablations, optimizer comparisons (DPO vs. GRPO), theoretical justification via LUPI, and full cost/procedural documentation. All additions are clearly marked with \added{}, including new pseudocode, tables, and dataset specifications.

We have also carefully revised wording, clarified the Generator’s role, corrected all noted typos, and strengthened baseline comparisons.

As the rebuttal period nears its end, we kindly ask if any further clarification would be helpful—we’re happy to provide it promptly.

Thank you again for your time, rigor, and constructive guidance.

Best regards,

Authors of submission 1985

---

### Author Response · Authors · 2025-12-02
**Summary of Contributions and Rebuttal**

Dear PCs, SACs, and ACs,

We sincerely thank the reviewers for their insightful feedback. We are encouraged by the positive recognition and grateful for the suggestions, which have significantly strengthened our paper. Below, we summarize the reviews and our extensive rebuttal efforts.

**Claim of Contribution**

We introduce Socratic-Zero, a fully autonomous framework that bootstraps reasoning from minimal seeds via the co-evolution of three agents. Our work addresses the data-dependency bottleneck by creating a closed-loop system where a Solver, Teacher, and Generator interact. Formalizing this as adaptive curriculum learning—where the Solver learns from preference feedback, the Teacher crafts challenges, and the Generator distills the Teacher's strategy for scalable, low-cost generation—we enable autonomous advancement without large-scale external datasets.

**Summary of Reviews and Responses**

We are glad reviewers appreciate our work's multi-faceted contributions, with **dpUA** highlighting its "substantial" results and "comprehensive" evaluation, and **K4xK** & **zAsq** praising the "novel" mechanism. We are encouraged that **one reviewer has already maintained a positive rating.**

The discussion has been constructive where possible. The other reviewers' reservations centered on requests for further empirical validation and clarification. Unfortunately, a platform technical issue has limited back-and-forth exchange with every reviewer since our major revisions. Nevertheless, their initial feedback indicated a thorough response, which we provide below, would be crucial for their final assessment.

We now address the specific points raised, all of which are carefully addressed in the table below.

| Reviewer | Reviewer's Concern/Questions | Author's Response |
| :--- | :--- | :--- |
| **dpUA, K4xK, zAsq, Db9h** | **Reliance on a vastly superior Teacher; effectiveness with weaker Teachers.** | We ran a full **Teacher-Solver scaling analysis** (New Table 12, App. R), proving **substantial gains (+10.5 pts) even with a weaker Teacher**. A new theoretical justification (App. S) explains *why* this is possible. |
| **XXnF, dpUA, Db9h** | **Generality of the framework beyond the initial domain.** | We **extended Socratic-Zero to multimodal geometric reasoning** (text+images). It achieved a +4.13 pt gain with **4x less data** than the top baseline, proving cross-modality generalization (New App. W). |
| **dpUA, K4xK, zAsq, Db9h**| **Scalability and the practical role of the Generator.** | A **new "Stage 4" experiment** (New Table 6) shows our low-cost Generator can replace the expensive Teacher with **nearly identical performance (60.5% vs. 60.3%)**, proving the framework's scalability and cost-effectiveness. |
| **dpUA, Db9h** | **Rigor of baseline comparisons.** | We added **three strong new baselines** (e.g., 75k distillation, 53k DPO). Socratic-Zero's outperformance remains significant, confirming its superiority over static, large-scale data augmentation (Updated Table 1). |
| **XXnF** | **Robustness of the "data-free" claim and dependency on initial seeds.** | A **"true from-zero" test using machine-generated seeds** yielded **near-identical performance (0.1% drop)**, confirming robustness and low dependency on seed quality (New App. V). |
| **dpUA** | **Potential bias from using the same model as enhancer and verifier.** | A **new verifier ablation** using different models shows **stable performance (≤0.4 pt drop)**, proving robustness against self-consistency bias (New App. T). |
| **XXnF** | **Justification for using DPO over alternatives like PPO.** | We ran a new **head-to-head comparison with a PPO variant (GRPO)**. DPO showed superior performance and efficiency, justifying our choice (New App. X). |
| **dpUA, zAsq** | **Lack of clarity on the overall training procedure and stage definitions.** | For full transparency, we added **detailed end-to-end pseudocode** (Alg. 2, App. Q) and **explicit stage definitions** (App. U), ensuring reproducibility. |
| **dpUA, XXnF** | **Need for detailed computational cost analysis.** | A **new cost analysis appendix** (App. P) quantitatively details the framework's efficiency and the practical benefits of the Generator replacement. |

These major revisions, driven by your feedback, have addressed all concerns and significantly elevated the paper's quality, rigor, and impact. We are confident it is now a more complete and impactful contribution.

We remain grateful for the opportunity to improve our paper and hope for continued dialogue.

Best regards,

The Authors of Submission 1985

---

### Meta-Review · Area_Chair_8oXw · 2026-01-05

**Summary:**

The paper introduces "Socratic-Zero," a fully autonomous framework designed to bootstrap the reasoning capabilities of LLMs using a minimal set of seed examples. The system relies on a co-evolutionary loop involving three agents: a Solver (learning via DPO), a Teacher (providing feedback and curriculum), and a Generator (distilling the curriculum strategy). The approach aims to solve the data scarcity problem by dynamically generating tailored problems without human annotation.

Despite the authors' strong rebuttal and the addition of 17 new experiments, the consensus remains that the paper's fundamental significance does not meet the bar for ICLR. Reviewers (e.g., zAsq) viewed the framework primarily as an engineering combination of established paradigms (Knowledge Distillation, DPO) rather than a methodological breakthrough, noting that the "co-evolution" narrative does not escape the theoretical bound set by the Teacher's capability. Furthermore, the system's high complexity—involving three distinct agents and multiple training stages—was criticized as "over-engineering"(Reviewers dpUA, XXnF), with a complexity-to-gain ratio that is difficult to justify compared to more streamlined self-improvement methods. Finally, critical theoretical limitations regarding the risk of misaligned guidance when the Solver inevitably surpasses the Teacher's verification ceiling (Reviewer K4xK) remain insufficiently addressed.

**Reviewer Concerns:**

Concerns Addressed by the Rebuttal:
•**Computational Cost**: Resolved by the new "Stage 4" experiment proving the lightweight Generator can replace the expensive Teacher.
•**Generalization**: Resolved by extending the framework to Multimodal Geometric Reasoning.
•**Teacher Dependence**: Addressed via scaling laws showing a weak Teacher (4B) can guide a strong Solver (8B).
•**Baseline Fairness**: Resolved by clarifying evaluation protocols (Zero-shot) and adding strong external baselines (Distillation-75k).

Outstanding Concerns:
•**Conceptual Novelty**: The contribution is seen as incremental—an advanced engineering application of distillation rather than a novel reasoning mechanism.
•**System Complexity**: The complexity-to-gain ratio is poor. Coordinating three agents is viewed as unnecessary "over-engineering" for practical adoption.
•**Verification Bottleneck**: The critical issue of how the system evolves once the Solver surpasses the Teacher's ability to verify correctness remains theoretically unaddressed.

**Reviewer Scores:**

I think the final scores are 4, 4, 6, 6, 4.

While Reviewer zAsq acknowledged that the rebuttal "effectively addressed his questions," he explicitly decided to "keep his original rating" of 6. Similarly, despite the extensive rebuttal efforts, the other reviewers (dpUA, XXnF, Db9h) did not raise their initial scores of 4, reflecting a remaining lack of consensus regarding the fundamental novelty and necessity of the proposed system complexity.

---

### Decision · Program_Chairs · 2026-01-26

Reject